# X-Field: A Physically Informed Representation for 3D X-ray Reconstruction

**Feiran Wang**[1*]    **Jiachen Tao**[1*]    **Junyi Wu**[1*]    **Haoxuan Wang**[1]    **Bin Duan**[2]
**Kai Wang**[3]    **Zongxin Yang**[4]    **Yan Yan**[1†]

[1]University of Illinois Chicago    [2]University of Michigan
[3]National University of Singapore    [4]Harvard Medical School

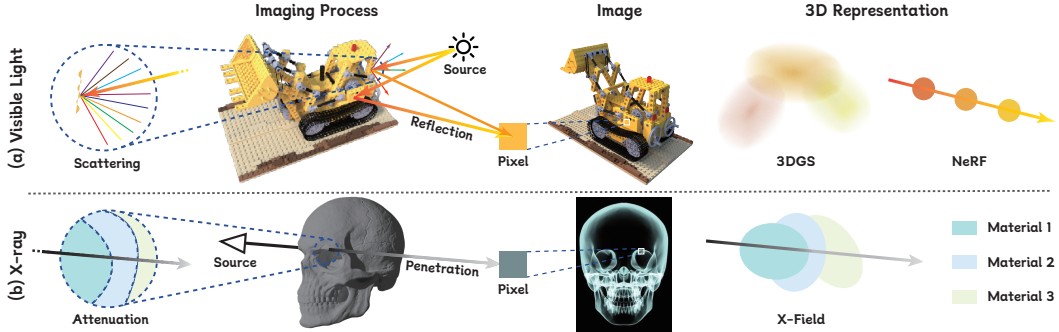

Figure 1: **Comparison of Imaging Processes and Corresponding 3D Representations.** (a) Visible light interacts with surfaces mainly through scattering and reflection. 3D representations of NeRF [1] and 3DGS [2] model this process by accumulating directional light rays. (b) Unlike previous approaches, our model represents internal structures using homogeneous ellipsoids to capture X-ray attenuation and penetration properties.

## Abstract

X-ray imaging is indispensable in medical diagnostics, yet its use is tightly regulated due to radiation exposure. Recent research borrows representations from the 3D reconstruction area to complete two tasks with reduced radiation dose: X-ray Novel View Synthesis (NVS) and Computed Tomography (CT) reconstruction. However, these representations fail to fully capture the penetration and attenuation properties of X-ray imaging as they originate from visible light imaging. In this paper, we introduce **X-Field**, a 3D representation informed in the physics of X-ray imaging. First, we employ homogeneous 3D ellipsoids with distinct attenuation coefficients to accurately model diverse materials within internal structures. Second, we introduce an efficient path-partitioning algorithm that resolves the intricate intersection of ellipsoids to compute cumulative attenuation along an X-ray path. We further propose a hybrid progressive initialization to refine the geometric accuracy of X-Field and incorporate material-based optimization to enhance model fitting along material boundaries. Experiments show that X-Field achieves superior visual fidelity on both real-world human organ and synthetic object datasets, outperforming state-of-the-art methods in X-ray NVS and CT Reconstruction. Our code is available on the project page: https://github.com/Brack-Wang/X-Field.

## 1   Introduction

X-ray imaging is a cornerstone of Computed Tomography (CT) reconstruction [3, 4, 5, 6], providing critical insights into internal structures for clinical diagnostics. X-rays undergo progressive attenuation

---

[*]Equal Contribution   [†]Corresponding Author

39th Conference on Neural Information Processing Systems (NeurIPS 2025).

while penetrating materials until the residual energy reaches the detector, forming an X-ray projection. Traditional CT reconstruction relies on hundreds of X-ray projections acquired from various angles to recover a density field [7, 8, 9]. However, acquiring such a vast number of projections exposes patients to high doses of ionizing radiation, posing health risks [10, 11]. Consequently, X-ray reconstruction has gained increasing attention [12, 13, 14, 15, 16, 17], aiming to synthesize novel X-ray views from a sparse set of input projections and facilitate the reconstruction of high-quality CT volumes.

Recent advances in 3D reconstruction [2, 1, 18] have laid the groundwork for X-ray reconstruction [14, 15, 16]. Yet, current 3D reconstruction techniques were originally designed for visible light imaging, as illustrated in Figure 1(a). When visible light rays interact with a surface, despite minor absorption, most wavelengths are *reflected* or *scattered* into multiple directions [19]. As a result, light rays from different directions can accumulate to form a single pixel, producing varying appearances of the same surface depending on the viewpoint. Neural Radiance Fields (NeRF) [1] employ deep networks to implicitly encode directional light information at each sampled point, conditioned by spatial positions and view directions to model appearance variation. Similarly, 3D Gaussian Splatting (3DGS) [2] projects anisotropic ellipsoids onto the image plane and accumulates the projected Gaussian splats to determine the pixel colors. Both methods reconstruct scenes considering the accumulation of multi-directional rays, thus well-suited for visible light imaging.

However, X-ray imaging operates on fundamentally different principles from visible light imaging, as depicted in Figure 1(b). First, X-rays are a form of high-energy electromagnetic radiation that is more capable of *penetrating* objects [20], whereas visible light, emitted from natural sources, has lower energy and is primarily reflected by surfaces. Second, X-rays progressively *attenuate* as they pass through an object in a nearly straight line before reaching the detector on the opposite side. Third, the pixel intensity is primarily determined by energy absorption from the corresponding X-ray, rather than by the combined contributions of visible light from multiple directions.

Therefore, X-ray imaging reveals materials composition with distinct densities distributed throughout the object's interior structure. As X-rays penetrate materials, their energy attenuates proportionally to both material density and path length through each substance. Recent X-ray reconstruction methods build upon 3DGS and achieve efficient imaging. For instance, X-Gaussian [15] replaces the spherical harmonics function with an intensity function to capture X-ray grayscale values, while $R^2$-Gaussian [16] extends this by correcting integration bias to enhance CT reconstruction quality. Yet, these methods represent each material as a Gaussian ellipsoid, depicted as 3DGS representation in Figure 1, with maximum density at the center that gradually diminishes toward the periphery, and render through ellipsoid superposition, without accounting for the critical factor of penetration length through each material. To faithfully adhere to the principles of X-ray imaging, X-ray reconstruction demands a representation that accurately models the radiological density across various materials.

In this paper, we present **X-Field**, a physically informed ellipsoid representation specifically designed to model X-ray attenuation as it penetrates materials during imaging. To differentiate various materials, X-Field represents an object's structure using ellipsoids with distinct attenuation coefficients, shown as X-Field representation in Figure 1. Then, to track the distance an X-ray travels through each material, we derive an explicit form for segment length.

For precise pixel intensity calculation, we express the attenuation integral along the ray path as a finite Riemann sum of the attenuation coefficients and segment lengths. Yet, during optimization, ellipsoids with different attenuation coefficients may overlap. To ensure that each position in 3D space corresponds to a single material, we further design a first-pass precedence strategy, resolving ellipsoid ambiguities in overlapping regions. Furthermore, we propose Material-Based Optimization, which adaptively refines ellipsoid placement by splitting them along material boundaries guided by the local attenuation gradient.

We evaluate X-Field on X-ray reconstruction across various modalities, including real-world human organ datasets and simulated object datasets. Furthermore, we compare X-Field with existing methods in sparse-view CT reconstruction based on Novel View Synthesis (NVS) results. Experiments demonstrate that X-Field outperforms state-of-the-art (SOTA) methods in both NVS and CT reconstruction. Notably, our method achieves optimal results within 10 minutes using only 10 input X-ray projections, delivering substantial PSNR improvements of 2.44 dB for X-ray NVS and 3.98 dB for CT reconstruction over SOTA methods.

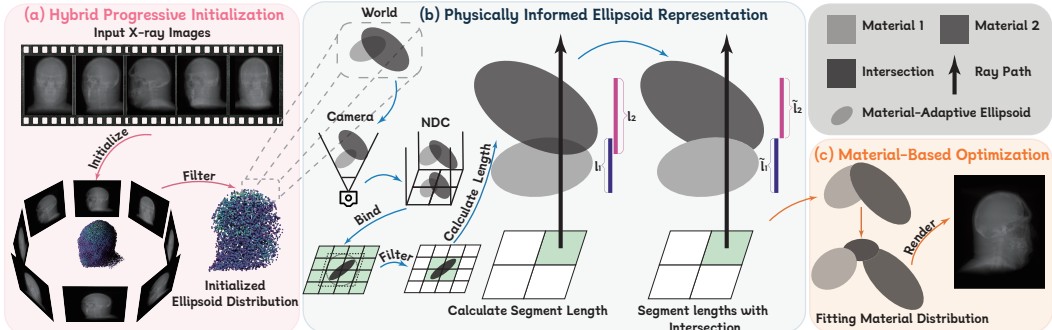

Figure 2: **Overview of X-Field.** (a) Hybrid Progressive Initialization. We begin with X-ray images to construct a coarse initialization using combined iterative methods. **(b) Physically Grounded Ellipsoid Representation.** We transform initialized ellipsoids into NDC space and associate them with pixels. We then compute the attenuation integral along the ray, considering segment length and ellipsoid intersections. (c) Material-Based Optimization. Our optimization captures detailed material boundaries for high-quality rendering.

## 2    Related Work

**X-ray Novel View Synthesis.**

3D X-ray reconstruction consists of two tasks: X-ray novel view synthesis and CT reconstruction [14]. X-ray NVS [14, 15, 13, 16] is essential for medical diagnostics, as it reconstructs images from sparse-view inputs, significantly reducing patient exposure to X-ray radiation. Motivated by the fact that visible light and X-rays are both electromagnetic waves, two representation methods originally designed for visible light have been adopted for X-ray reconstruction: Neural Radiance Fields (NeRF) [1], which employ multi-layer perceptrons for implicit encoding, and 3D Gaussian Splatting (3DGS) [2], which uses Gaussian ellipsoids for explicit representation. NeRF-based methods, such as NAF [13], improve efficiency by incorporating hash tables, while SAX-NeRF [14] introduces a transformer architecture to better model 3D structural dependencies.

3DGS-based approaches, such as X-Gaussian[15], replace the spherical harmonics color in 3DGS with a response function, addressing differences between visible light color and X-ray intensity. $R^2$-Gaussian[16] further corrects integration bias for more accurate density retrieval. However, neither representation has a comprehensive consideration of the attenuation and penetration properties of the X-ray. Instead, we introduce attenuation-adaptive ellipsoids with distinct attenuation coefficients and consider the X-ray penetration length within the materials, ensuring more accurate reconstructions.

**Computed Tomography Reconstruction.** CT reconstruction is of vital importance for domains such as medical diagnosis [6, 21, 22, 5, 23, 24], biology [25, 26, 27], industrial inspection [28, 29], and security screening [30]. Early CT reconstruction methods [20] can be categorized into analytical approaches [3, 31] and iterative techniques [4, 32, 33]. Analytical methods such as FDK [3] perform a filtered back-projection of acquired projections. Iterative methods such as SART [4] and CGLS [34] reconstruct CT images by iteratively refining the solution through algebraic projection corrections and solving a least-squares optimization problem, respectively. However, these methods typically require hundreds to thousands of X-ray images, leading to increased radiation exposure.

Fortunately, X-ray NVS offers an effective alternative to CT reconstruction. By generating diverse novel views from sparse X-ray inputs [15, 16, 14, 17, 35], it enables reconstruction of high-quality CT volumes from limited data. We further reduce the number of required X-ray images to as few as five while achieving superior performance both qualitatively and quantitatively compared to other baselines, highlighting its practicality for real-world applications.

## 3    Method

In this section, we introduce our Physically Informed Ellipsoid Representation in Sec. 3.1, including (1) the formulation of X-ray Physical Field, (2) the Attenuation-Adaptive Ellipsoids, (3) the algorithms for calculating segment lengths with intersections, and (4) the overlap filtering method. Then, we propose Hybrid Progressive Initialization in Sec. 3.2 and Material-Based Optimization in Sec. 3.3.

### 3.1 Physically Informed Ellipsoid Representation

**X-ray Physical Field.** X-ray imaging quantifies the cumulative attenuation of X-rays as they traverse an object, governed by the material-dependent absorption properties of the medium. To formally describe this process, we define the X-ray Physical Field, where each spatial position $\mathbf{x} \in \mathbb{R}^3$ is characterized by the local energy absorption rate $\sigma(\mathbf{x}) \in \mathbb{R}^+$ of X-ray [20, 5, 36]. When X-rays with an initial intensity $I_0$ propagate through the X-ray Physical Field, their energy is progressively attenuated by the materials they traverse. Ultimately, the remaining intensity forms a projection image $I \in \mathbb{R}^{H \times W}$. Mathematically, we represent an X-ray path as $\mathbf{r}(t) = \mathbf{o} + t\mathbf{d} \in \mathbb{R}^3$, where $\mathbf{o}$ denotes the X-ray source position, $\mathbf{d}$ is the unit view direction vector, and $t$ varies between the entry $t_0$ and exit $t_n$ points of the object. According to the Beer-Lambert law [37], the X-ray intensity $I'(\mathbf{r})$ after attenuation is given by:

$$I'(\mathbf{r}) = I_0 \exp\left(-\int_{t_0}^{t_n} \sigma(\mathbf{r}(t))\, dt\right). \tag{1}$$

In practice, the raw data is typically processed in logarithmic space for computational efficiency [20]:

$$I(\mathbf{r}) = \log I_0 - \log I'(\mathbf{r}) = \int_{t_0}^{t_n} \sigma(\mathbf{r}(t))\, dt. \tag{2}$$

Thus, each pixel intensity $I(\mathbf{r})$ in the projection image aggregates the material absorption along the X-ray path, providing the foundation for reconstructing internal structures.

**Attenuation-Adaptive Ellipsoids.** Inspired by the infinitesimal method [38], we represent the spatial distribution of materials using a set of $N$ ellipsoids $\{\mathbf{E}_i\}$. Each ellipsoid $\mathbf{E}_i$ encodes material properties and contains a center position $\mathbf{p_c}$ and covariance matrix $\mathbf{\Sigma}_{3D}$. To maintain physical consistency with X-ray imaging, we further assign a distinct, non-negative attenuation coefficient $\sigma_i$ to characterize the local absorption behavior. Given an X-ray path $\mathbf{r}$ that traverses multiple ellipsoids, the corresponding pixel intensity is determined by the accumulated attenuation along the path:

$$
\begin{aligned}
I(\mathbf{r}) = \int_{t_0}^{t_n} \sigma(\mathbf{r}(t))\, dt &= \int_{t_0}^{t_1} \sigma(\mathbf{r}(t))\, dt + \cdots + \int_{t_{n-1}}^{t_n} \sigma(\mathbf{r}(t))\, dt \\
&= \sigma_0 \int_{t_0}^{t_1} dt + \cdots + \sigma_{n-1} \int_{t_{n-1}}^{t_n} dt \\
&= \sigma_0 l_0 + \sigma_1 l_1 + \cdots + \sigma_{n-1} l_{n-1},
\end{aligned}
\tag{3}
$$

where $l_i = t_{i+1} - t_i \geq 0$ represents the segment length of the X-ray path within ellipsoid $\mathbf{E}_i$. Therefore, accurately measuring the accumulated attenuation requires precise determination of each segment length $l_i$. In the following, we derive an explicit formulation for $l_i$.

**Explicit Form of Segment Lengths.** Our objective is to compute the segment length $l_i$ for ellipsoid $\mathbf{E}_i$ along the view direction $\mathbf{d}$, a process illustrated in Figure 2(b). To achieve this, we first transform $\mathbf{E}_i$ from world to camera coordinates and then into Normalized Device Coordinates (NDC) space, aligning X-rays with the coordinate axes for computational simplicity. To establish a reference for general segment length computation, we define the maximum segment length $l_{\max}$ as a pivot, which corresponds to the ray path passing through the center of $\mathbf{E}_i$:

$$l_{\max} = \frac{2}{\sqrt{\mathbf{d}^\top \mathbf{\Sigma}_{3D}^{-1} \mathbf{d}}}. \tag{4}$$

Given $l_{\max}$, we then compute the segment length $l_i$ for an arbitrary point $\mathbf{u}$ inside the projected ellipse of $\mathbf{E}_i$ in NDC space. The relationship is formulated as:

$$l_i = l_{\max} \times \sqrt{1 - \left(\frac{C - B^2}{A}\right)}, \quad \text{where} \tag{5}$$

$$A = \mathbf{d}^\top \mathbf{\Sigma}_{3D}^{-1} \mathbf{d}, \quad B = \mathbf{a}^\top \mathbf{\Sigma}_{3D}^{-1} \mathbf{d}, \quad C = \mathbf{a}^\top \mathbf{\Sigma}_{3D}^{-1} \mathbf{a}. \tag{6}$$

Here, $\mathbf{a} = \mathbf{u} - \mathbf{p_c}$ is the displacement vector from the center of the ellipsoid $\mathbf{p_c}$ to the point $\mathbf{u}$. The full derivation of $l_{\max}$ and $l_i$ is provided in the supplementary materials.

**Segment Lengths with Intersections.** As ellipsoids adjust their positions and sizes during optimization, intersections may occur, leading to ambiguities in the attenuation rates within overlapping regions. To ensure that each location corresponds to a single dominant material, resolving these complex overlaps is crucial to maintaining a physically consistent representation.

For optimization stability, we adopt a simple yet effective first-pass precedence strategy. Given two intersecting ellipsoids along the X-ray path, we first sort them in ascending depth order, denoted by $\mathbf{E}_i$ and $\mathbf{E}_{i+1}$, where $\mathbf{E}_i$ appears first along the ray. The overlapping region $e_i$ is then exclusively assigned to $\mathbf{E}_i$ by adjusting the effective region of $\mathbf{E}_{i+1}$ to $\mathbf{E}_{i+1} - e_i$. This modification directly impacts the segment length computation, requiring additional updates to ensure consistency in attenuation accumulation, as shown in Figure 2(b). To systematically handle different intersection scenarios, we introduce an efficient correction function $f$, as outlined in Algorithm 1. The final accumulated attenuation along the X-ray path is then expressed as:

---

**Algorithm 1** Segment Length Correction with Intersections

---

**Input:** $(z_0, z_1, \ldots, z_{n-1})$: Sorted depths
  $(l_0, l_1, \ldots, l_{n-1})$: Initial segment lengths
**Output:** Updated segment lengths $\tilde{l}_0, \tilde{l}_1, \ldots, \tilde{l}_{n-1}$
1:  **for** $i = 0$ **to** $n - 1$ **do**
2:    **if** $i == 0$ **then**
3:      $\tilde{l}_0 \leftarrow l_0$
4:      $z \leftarrow z_0, l \leftarrow l_0$
5:    **else**
6:      **if** $z_i < z + \frac{1}{2}l$ **then**
7:        $\tilde{l}_i \leftarrow \max(0, (z_i + \frac{1}{2}l_i) - (z + \frac{1}{2}l))$
8:      **else**
9:        $\tilde{l}_i \leftarrow \min(l_i, (\frac{1}{2}l_i + z_i) - (z + \frac{1}{2}l))$
10:     **end if**
11:     **if** $\tilde{l}_i \neq 0$ **then**
12:       $[z_i + \frac{1}{2}l_i - \tilde{l}_i, z_i + \frac{1}{2}l_i]; z \leftarrow z_i, l \leftarrow l_i$
13:     **end if**
14:   **end if**
15: **end for**

---

$$I(\mathbf{r}) = \sum_{i=0}^{n-1} \sigma_i \cdot f\left(\frac{2}{\sqrt{A}} \times \sqrt{1 - \left(\frac{C - B^2}{A}\right)}\right), \tag{7}$$

**Physically Faithful Overlap Filtering.** While resolving segment length inconsistencies ensures a physically valid attenuation model along the X-ray path, accurately associating ellipsoids with their corresponding pixels is equally critical for maintaining consistency in the projected space. Current methods often rely on coarse bounding-box approximations, leading to unintended pixel assignments that compromise reconstruction fidelity. To address this, we introduce a refined Oriented Bounding Box (OBB) strategy that establishes precise pixel-ellipsoid associations. As illustrated in Figure 3, existing approaches [2, 15, 16] typically rely on Axis-Aligned Bounding Boxes (AABB) [39], which approximate the projected ellipse using a circumscribed circle. However, this method includes extraneous pixels (red region) that do not truly overlap with the projected ellipse, leading to physically inconsistent assignments. In contrast, we first determine the OBB via eigenvalue decomposition, aligning it with the principal axes of the projected ellipse. Then, we filter out irrelevant pixels while preserving only the valid tiles (green region) that are closely related to the actual projected ellipse. By eliminating spurious pixel association, our method enhances the physical consistency and numerical stability of the reconstruction process.

### 3.2 Hybrid Progressive Initialization

It is widely acknowledged that initializing point clouds with detailed geometric information accelerates model convergence [2, 15, 40]. However, conventional initialization techniques fail in the context of X-ray imaging. Specifically, SfM-based methods [41, 2] estimate point clouds by feature matching across multiple images. While in X-ray imaging, variations in ray paths significantly alter the remaining energy intensity, leading to feature mismatches and unreliable correspondences. Similarly, point cloud regression methods such as Dust3R [42, 43, 44] directly infer 3D structures from images but fail in X-ray reconstruction due to domain shifts, as they are trained on natural images and do not account for X-ray-specific attenuation properties.

To address this challenge, we design a Hybrid Initialization strategy that progressively refines the geometric structure through a sequence of iterative methods. We compare the results in Figure 4. Our process begins with Conjugate Gradient Least Squares (CGLS) [45, 34, 20], which efficiently

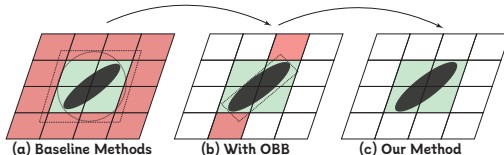

Figure 3: **Pixel-Ellipse Association.** (a) AABB [39] results in incorrect associations. (b) OBB [47] pixels. (c) Ours keeps only aligned pixels.



Figure 4: **Initialization Comparisons.** Ours produces informative geometry prior, compared to Random of X-Gaussian [15] and FDK of $R^2$-Gaussian [16].

provides a coarse global estimate. Next, we refine this estimate using Simultaneous Algebraic Reconstruction Technique (SART) [4], leveraging its capability to enhance local details and correct early-stage inconsistencies. Finally, we incorporate Total Variation (TV) regularization [46] into the initialization process to suppress noise and artifacts, preserving critical structural edges. For each 3D point obtained from our initialization, we define an ellipsoid centered at that position, assigning it a random covariance matrix $\Sigma_{3D}$ and an attenuation coefficient $\sigma_i$. We subsequently devise an optimization method for these parameters for better modeling the detailed material distribution.

## 3.3 Material-Based Optimization

Mainstream optimization strategies [2, 15, 16, 40] involve splitting and cloning in regions with poor structural fidelity while pruning with negligible opacity. However, this geometry-based optimization fails to capture the material composition along materials' edge. As depicted in Figure 5(a), the ground-truth object consists of three distinct materials, represented by different colors. While the geometry-based optimization successfully reconstructs the overall shape, it captures only two intermediate material types, as shown in Figure 5(b), highlighting the limitations of purely geometry-centric approaches in faithfully modeling material compositions.

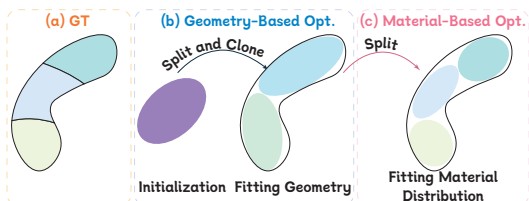

Figure 5: **Illustration of Adaptive Optimization Strategy.** (a) Ground-truth Material Distribution with different colors indicating distinct materials. (b) Geometry-Based Optimization [2], which fits the ellipsoids closely to the object geometry. (c) Our Material-Based Optimization, which further refines the ellipsoids to capture the material distribution.

The challenge arises from the continuous fluctuation of an ellipsoid's attenuation coefficient, which hinders stable convergence and prevents accurate material differentiation. To address this, we propose a material-based optimization strategy that explicitly accounts for the heterogeneous material distributions inherent in the X-ray Physical Field. Our approach is designed to identify material boundaries through local density estimation. Concretely, we randomly sample a subset of ellipsoids and compute the k-nearest neighbors [48], averaging their distances and analyzing the attenuation gradient. Regions with high density and steep gradients indicate material transitions, necessitating a finer adjustment. Therefore, to enhance fidelity, we selectively split ellipsoids in these complex regions, scaling each by a factor of 1.6, which follows empirical heuristics in prior studies [2, 15, 16]. As presented in Figure 5(c), our strategy significantly improves material separation, faithfully capturing the internal composition of objects.

## 4 Experiment

### 4.1 Experiment Settings

**Dataset.** Following the literature convention [14, 15], we conduct experiments on a large-scale X3D dataset containing 15 scenes with two collections: Human Organs, derived from real-world medical datasets, to evaluate model performance in the medical domain; and Daily Objects, generated from synthetic datasets, to assess generalization ability. Specifically, chest scans are sourced from LIDC-IDRI [49], pancreas CT scans from Pancreas-CT [50]. The remaining objects are obtained from VOLVIS [51] and the open scientific visualization dataset [52]. We adopt the tomography toolbox TIGRE [53] to capture projections from CT volumes in the range of $0° \sim 180°$ with minor scatter and electronic noise. For highly sparse-view novel view synthesis, 5 and 10 views are used for

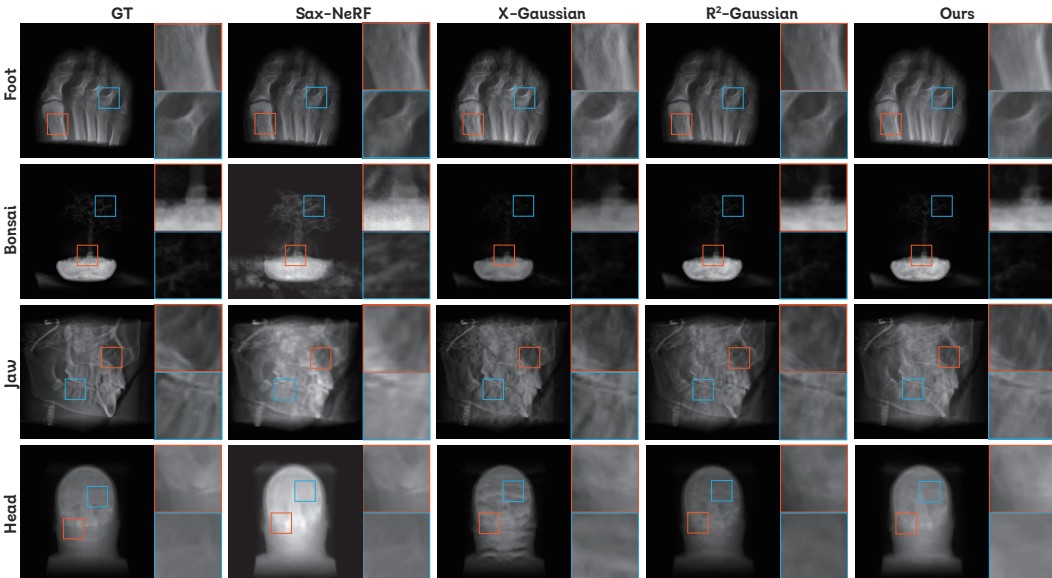

Figure 6: **Qualitative Comparison of NVS.** We present visual examples of reconstructed images across four cases trained with 10 views. Our results demonstrate superior visual quality, richer details, and fewer artifacts.

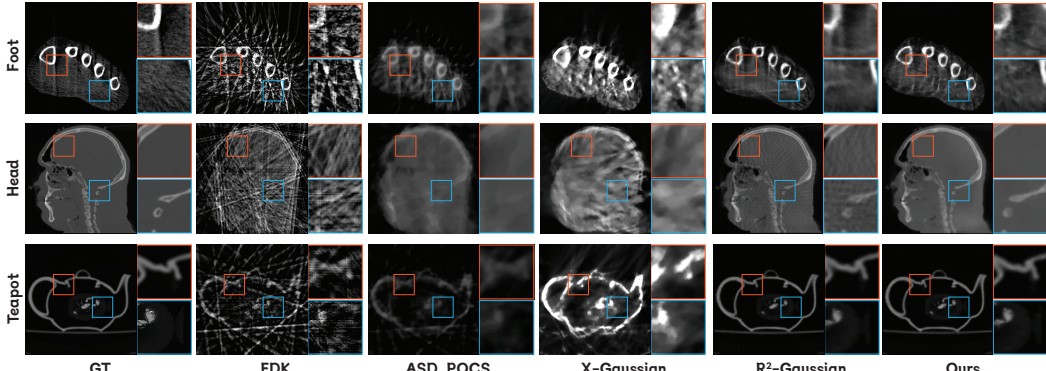

Figure 7: **Qualitative Comparison of CT Reconstruction.** Our method produces clearer textures, more refined anatomical structures, and fewer artifacts, particularly in high-contrast regions such as the cranial cavity.

training and 50 views are used for testing. To further assess model performance and scalability, we generate 50, 25, and 15 views for evaluating the performance under sparse-view synthesis.

**Baselines.** We compare X-Field with state-of-the-art 3D X-ray reconstruction methods, including TensoRF [12], NeAT [17], NAF [13], SAX-NeRF [14], X-Gaussian [15], and $R^2$-Gaussian [16]. TensoRF, NeAT, NAF, and SAX-NeRF are NeRF-based methods designed for efficient reconstruction, with SAX-NeRF achieving SOTA performance among them by incorporating a transformer architecture as the model backbone. X-Gaussian and $R^2$-Gaussian are 3DGS-based methods, where X-Gaussian focuses on NVS, and $R^2$-Gaussian extends its applicability to CT reconstruction by introducing voxelization. We also compare against traditional CT reconstruction methods including FDK [3] and SART [4]. The novel view images are obtained by leveraging TIGRE for rendering.

**Metrics.** We adopt peak signal-to-noise ratio (PSNR) [54] to assess the quality of rendered images, structural similarity index measure (SSIM) [55] to measure consistency between predicted images and ground-truth, and Learned Perceptual Image Patch Similarity (LPIPS) [56] to analyze the perceptual quality in high-level feature space. For clarity, we report LPIPS as LPIPS* = LPIPS $\times 10^3$ instead.

**Implementation Details.** For X-ray novel view synthesis, we evaluate learning-based methods using their official implementations. For CT reconstruction, we follow [17, 14, 15] to synthesize novel-view X-ray images from sparse inputs and reconstruct CT volumes using a total of 100 images. All experiments were conducted on a single RTX 6000 Ada GPU.

| Method | Human Organ 10-views [16] | | | Daily Object 10-views [16] | | | Human Organ 5-views [16] | | | Daily Object 5-views [16] | | |
|---|---|---|---|---|---|---|---|---|---|---|---|---|
| | PSNR↑ | SSIM↑ | LPIPS*↓ | PSNR↑ | SSIM↑ | LPIPS*↓ | PSNR↑ | SSIM↑ | LPIPS*↓ | PSNR↑ | SSIM↑ | LPIPS*↓ |
| *Traditional Methods* | | | | | | | | | | | | |
| FDK [3] | 12.35 | 0.675 | 291.2 | 16.52 | 0.716 | 259.1 | 8.15 | 0.618 | 310.6 | 14.42 | 0.688 | 283.7 |
| SART [4] | 13.23 | 0.691 | 284.8 | 17.69 | 0.724 | 247.3 | 9.31 | 0.634 | 303.4 | 15.68 | 0.663 | 293.5 |
| *Deep Learning-based Methods* | | | | | | | | | | | | |
| TensoRF [12] | 16.61 | 0.928 | 182.5 | 24.19 | 0.946 | 153.4 | 12.32 | 0.895 | 189.6 | 18.27 | 0.922 | 210.8 |
| NeAT [17] | 16.22 | 0.934 | 185.3 | 25.15 | 0.957 | 155.2 | 11.08 | 0.887 | 188.3 | 17.29 | 0.908 | 211.3 |
| NAF [13] | 17.89 | 0.925 | 193.2 | 25.44 | 0.949 | 151.9 | 11.19 | 0.894 | 197.1 | 17.02 | 0.923 | 208.5 |
| SAX-NeRF [14] | 19.32 | 0.945 | 186.4 | 25.38 | 0.979 | 143.6 | 14.18 | 0.901 | 191.2 | 19.09 | 0.948 | 204.9 |
| X-Gaussian [15] | 22.88 | 0.947 | 130.3 | 22.91 | 0.982 | 79.12 | 17.23 | 0.947 | 176.4 | 20.31 | 0.961 | 108.1 |
| $R^2$-Gaussian [16] | 33.72 | 0.967 | 85.97 | 41.93 | 0.986 | 54.31 | 31.12 | 0.956 | 109.7 | 34.52 | 0.965 | 82.46 |
| **Ours** | 35.71 | 0.980 | 71.03 | 42.80 | 0.983 | 45.64 | 32.34 | 0.963 | 103.2 | 37.41 | 0.970 | 81.02 |

Table 1: **Quantitative Comparison of NVS.** We compare our X-Field with: (a) Traditional analytical method: FDK , SART . (b) Deep Learning-based methods: TensoRF , NeAT , NAF , SAX-NeRF , X-Gaussian , and $R^2$-Gaussian . We report LPIPS* = LPIPS $\times 10^3$. We mark out `best` and `second best` method for all metrics.

## 4.2 Comparison with State-of-the-Art Methods

We present the quantitative results of X-ray NVS and CT reconstruction, discussing the qualitative results across all scenes, highlighting the superior performance of X-Field.

**Discussion on NVS Quantitative Results.** We compare X-Field with two traditional methods (FDK, SART), three NeRF-based methods (TensoRF, NeAT, NAF), and three SOTA methods (SAX-NeRF, X-Gaussian, and $R^2$-Gaussian). Table 1 reports the quantitative results of highly sparse views (10 views and 5 views) X-ray NVS. Note that we report quantitative results as the mean results across all scenes under the same setting, and scene-wise results are presented in the supplementary material. X-Field demonstrates superior performance in reconstructing X-ray novel views across most scenarios, surpassing the state-of-the-art $R^2$-Gaussian in all metrics. Specifically, in the human organ reconstruction setting, X-Field consistently outperforms the state-of-the-art $R^2$-Gaussian, achieving higher SSIM scores and competitive PSNR and LPIPS values, highlighting its effectiveness for complex reconstruction scenarios.

**Discussion on NVS Qualitative Results.** Figure 6 presents visual comparisons between X-Field and multiple methods, including SAX-NeRF, X-Gaussian, and $R^2$-Gaussian. These highly sparse view settings provide limited information, resulting in artifacts of varying severity across all methods. SAX-NeRF reconstructs the overall structure but introduces noticeable blurry artifacts, particularly in the bonsai scene. X-Gaussian produces line and wave-pattern artifacts, which are prominent in the head scene. While $R^2$-Gaussian performs better than the other baselines, it exhibits flaws in the fine details of the bone structure. For example, $R^2$-Gaussian introduces black linear artifacts in the bone region in the foot scene, where ours produces smoother textures. In summary, our method is able to effectively mitigate blurry, line, and wave-pattern artifacts while maintaining smoothness in distinct areas and on object surfaces. More visual comparisons are provided in supplementary video.

**Discussion on CT Reconstruction Results.** Following the tradition [14, 15, 16], we also compare X-Field with traditional learning-free algorithms, FDK [3] and ASD_POCS [57], as well as learning-based methods including X-Gaussian [15] and $R^2$-Gaussian [16] for sparse-view CT reconstruction. Specifically, we assess the performance of 3DGS-based methods and X-Field by generating novel view images from sparse input projections (5 and 10 views) and reconstructing CT scans using ASD_POCS with a total of 100 views. The quantitative results, presented in Table 2, demonstrate that X-Field consistently achieves the best performance across all scenarios. In particular, when generating novel X-ray projections from 10 input views, X-Field combined with ASD_POCS achieves a PSNR of 33.26 and SSIM of 0.911, a comparable result with the SOTA method, $R^2$-Gaussian.

Figure 7 shows qualitative results of sparse-view CT reconstruction of foot, head, and teapot scans using 10 input views. Without novel view projections, FDK introduces streak artifacts, while ASD_POCS results in blurred structural details. The NVS of X-Gaussian produces clearer reconstructions but introduces shadow artifacts. In contrast, our method and $R^2$-Gaussian achieve superior visual quality, with X-Field delivering better results by recovering smoother textures and reducing needle-like artifacts, especially in delicate structures such as the cranial region.

## 4.3 Ablation Study

To comprehensively assess the performance of X-Field, we evaluate the impact of the proposed components, compare different initialization strategies, and evaluate X-Field under various input view settings from 5 to 50 views.

| #Views | 5 | | 10 | |
|---|---|---|---|---|
| Method | PSNR ↑ | SSIM ↑ | PSNR ↑ | SSIM ↑ |
| FDK | 15.20 | 0.144 | 19.95 | 0.257 |
| ASD_POCS | 24.89 | 0.731 | 27.50 | 0.787 |
| X-Gaussian | 17.57 | 0.688 | 18.44 | 0.537 |
| $R^2$-Gaussian | 26.83 | 0.804 | 29.28 | 0.946 |
| **Ours** | 28.04 | 0.815 | 33.26 | 0.911 |

Table 2: **CT Reconstruction Comparison**. Ours achieve comparable results under 5 and 10 views.

| Method | PSNR ↑ | SSIM ↑ | LPIPS* ↓ |
|---|---|---|---|
| w/o Material Opt. | 34.78 | 0.941 | 73.45 |
| w/o Overlap Filter | 34.59 | 0.937 | 74.32 |
| w/o Intersection | 33.84 | 0.929 | 76.60 |
| w/o Segment Length | 27.48 | 0.875 | 87.83 |
| **Ours** | 35.03 | 0.953 | 72.12 |

Table 3: **Ablation on Components**. Best performance when all components included.

| Initialization | PSNR ↑ | SSIM ↑ | LPIPS* ↓ |
|---|---|---|---|
| Random | 37.85 | 0.966 | 60.02 |
| FDK [3] | 37.96 | 0.967 | 59.81 |
| **Ours** | 38.67 | 0.969 | 59.95 |

Table 4: **Initialization Methods.** Our initialization improves reconstruction accuracy.

| #Views | PSNR ↑ | SSIM ↑ | LPIPS* ↓ |
|---|---|---|---|
| 5 | 31.61 | 0.933 | 92.23 |
| 10 | 35.11 | 0.959 | 83.72 |
| 25 | 41.71 | 0.979 | 75.26 |
| 50 | 42.61 | 0.993 | 61.19 |

Table 5: **Number of Input Views.** Performance improves as input views increase.

**Component Analysis.** Table 3 evaluates the effects of individual components on reconstruction performance. We observe that Material-Based Optimization and Overlap Filter have a limited impact on reconstruction quality. Removing the Intersection Module (Sec. 3.1) leads to substantial performance drops, with a PSNR drop of 1.19 dB, and an SSIM decrease of 0.024, underscoring its importance in preserving structural integrity.

Segment Length is a fundamental component of our ellipsoid representation, defined in Sec. 3.1 for capturing the distance each ray traverses within the ellipsoid. Removing it also severely impacts the model's ability to render novel views, leading to significant performance degradation, with a 7.6 dB drop in PSNR and an 11.71 increase in LPIPS. These results underscore the importance of Segment Length in enabling X-Field to achieve high-quality reconstructions.

**Initialization Analysis** We compare our hybrid initialization with random initialization used in X-Gaussian [15] and FDK [3] introduced in $R^2$-Guassian [16]. As shown in Figure 4, ours provides a point cloud with clearer boundaries and reduced noise, facilitating initialization. Table 4 shows that both FDK and our hybrid initialization outperform random initialization. While FDK achieves a slightly lower LPIPS, it also results in lower PSNR and SSIM. In contrast, ours improves both PSNR and SSIM, demonstrating its effectiveness in enhancing reconstruction quality.

**Input View Number Analysis.** To further demonstrate the scalability of X-Field, we conduct experiments to assess the effect of the input view number on reconstruction performance. As shown in Table 5, as the number of input views increases, the performance is consistently enhanced. When using 50 views as input, X-Field achieves performance comparable to work designed for sparse view X-Ray reconstruction [15, 14]. We further show in Figure 8 that, with the increase in the input number, the synthesis X-ray images exhibit smoother and clearer bone textures.

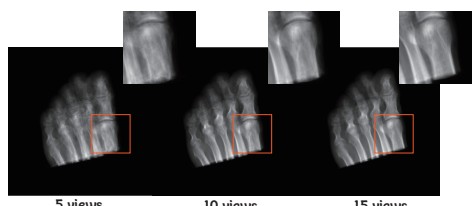

Figure 8: **Input View Numbers.** Five views enable good NVS, and additional input views enhance texture smoothness.

## 5 Conclusion

We present X-Field, a physically informed representation derived from the penetration and attenuation properties of X-rays for X-ray imaging. To model diverse materials within the internal structures of objects, we introduce attenuation-adaptive ellipsoids with distinct attenuation coefficients. To compute pixel intensity, we design a segment length algorithm that incorporates ellipsoid intersections, enabling accurate estimation of each material's X-ray energy absorption. To improve reconstruction performance, we optimize the ellipsoids along material boundaries and refine the geometric properties during initialization. Our proposed X-Field significantly outperforms state-of-the-art methods in both X-ray NVS and CT reconstruction, demonstrating strong potential for medical applications. Furthermore, we provide insights into the fundamental design of X-ray imaging, which can be extended to other tasks such as the reconstruction of translucent objects.

**Acknowledgments:** This research is supported by NSF IIS-2525840, CNS-2432534, ECCS-2514574, NIH 1RF1MH133764-01 and Cisco Research unrestricted gift. This article solely reflects opinions and conclusions of authors and not funding agencies.

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

## Technical Appendices and Supplementary Material

## Overview

## A    Detailed Derivations for Segment Lengths

In this work, we define the ray space using normalized device coordinates (NDC) [16], providing a convenient framework for deriving explicit segment length formulas. In this space, the segment length $l_i$ for each ellipsoid $\mathbf{E}_i$ is calculated along the view direction $\mathbf{d}$ of a given ray path. Leveraging the orthographic property of NDC space, the 3D ellipsoid $\mathbf{E}_i$ can be projected onto a 2D plane, forming an ellipse. The segment lengths are then constrained within this 2D ellipse, with values outside the ellipse set to zero.

Our goal is to find the relationship between the segment length and the point $\mathbf{u} = (x, y, 0)$ on the ellipse traversed by the ray. Below, we first derive the special value $l_{\max}$, which is the maximum distance a ray travels through the ellipsoid when originating from the center of the ellipse [58]. Then, we use $l_{\max}$ as a bridge to express $l_i$ for rays originating from other points on the ellipse.

Consider a ray originating from the center of the ellipse $\mathbf{u_c}$ and traveling along direction $\mathbf{d}$. The modeling of the ray is: $\mathbf{R}(s) = \mathbf{u_c} + s\mathbf{d} \in \mathbb{R}^3$, where $s$ is a scalar parameter. Meanwhile, the ellipsoid centered at $\mathbf{p_c}$ can be described as

$$(\mathbf{p} - \mathbf{p_c})^\top \mathbf{\Sigma}_{3D}^{-1} (\mathbf{p} - \mathbf{p_c}) = 1. \tag{8}$$

Substituting $\mathbf{p} = \mathbf{R}(s)$ into above equation, we have $s$ as in

$$(\mathbf{u_c} + s\mathbf{d} - \mathbf{p_c})^\top \mathbf{\Sigma}_{3D}^{-1} (\mathbf{u_c} + s\mathbf{d} - \mathbf{p_c}) = 1. \tag{9}$$

In the NDC space, translating an object does not affect its geometric length. Therefore, without loss of generality, we can assume that $\mathbf{p_c}$ lies on $\mathbf{u_c}$, and then we have:

$$(\mathbf{u_c} + s\mathbf{d} - \mathbf{p_c})^\top \mathbf{\Sigma}_{3D}^{-1} (\mathbf{u_c} + s\mathbf{d} - \mathbf{p_c}) = s^2 \mathbf{d}^\top \mathbf{\Sigma}_{3D}^{-1} \mathbf{d} = 1. \tag{10}$$

Solving this simplified equation, we can obtain the roots:

$$s = \pm \frac{1}{\sqrt{A}}, \quad \text{where} \quad A = \mathbf{d}^\top \mathbf{\Sigma}_{3D}^{-1} \mathbf{d}. \tag{11}$$

The segment length corresponding to $\mathbf{u_c}$ is the difference between the two roots: $l_{\max} = |s_2 - s_1| = \frac{2}{\sqrt{A}}$.

Next, we consider a general point $\mathbf{u}$ on the ellipse plane. The ray originating from $\mathbf{u}$ in the direction $\mathbf{d}$ can be represented by $\mathbf{R}(s) = \mathbf{u} + s\mathbf{d}$. Similarly, we define $\mathbf{a} = \mathbf{u} - \mathbf{p_c}$ and substitute $\mathbf{R}(s)$ into the ellipsoid equation:

$$(\mathbf{a} + s\mathbf{d})^\top \mathbf{\Sigma}_{3D}^{-1}(\mathbf{a} + s\mathbf{d}) = 1. \tag{12}$$

Expanding the equation results in:

$$s^2 \mathbf{d}^\top \mathbf{\Sigma}_{3D}^{-1}\mathbf{d} + 2s\mathbf{a}^\top \mathbf{\Sigma}_{3D}^{-1}\mathbf{d} + \mathbf{a}^\top \mathbf{\Sigma}_{3D}^{-1}\mathbf{a} = 1. \tag{13}$$

For simplicity, we denote $B = \mathbf{a}^\top \mathbf{\Sigma}_{3D}^{-1}\mathbf{d}$ and $C = \mathbf{a}^\top \mathbf{\Sigma}_{3D}^{-1}\mathbf{a}$. Thus, the equation becomes $As^2 + 2Bs + C - 1 = 0$, whose roots are:

$$s_{1,2} = \frac{-B \pm \sqrt{B^2 - A(C-1)}}{A}. \tag{14}$$

The segment length can then be derived as:

$$l_i = |s_2 - s_1| = l_{\max} \times \sqrt{1 - \left(\frac{C - B^2}{A}\right)}. \tag{15}$$

## B  3DGS Preliminaries

3D Gaussian Splatting (3DGS) [2] represents a scene as a collection of anisotropic Gaussian kernels in 3D space, denoted as $\mathbb{G}^3 = \{G_i^3\}_{i=1}^M$. Each Gaussian is parameterized by its center position, covariance matrix, color, and opacity. By splatting these kernels onto the image plane using a differentiable rasterizer, 3DGS enables efficient view synthesis and supports end-to-end optimization.

Formally, an RGB image $\mathbf{I}_{rgb} \in \mathbb{R}^{H \times W \times 3}$ is rendered from the 3D Gaussians as

$$\mathbf{I}_{rgb} = \mathcal{R}(\mathbb{G}^3) = \mathcal{C}\big(\mathcal{P}(\mathcal{T}(\mathbb{G}^3))\big), \tag{16}$$

where $\mathcal{T}$, $\mathcal{P}$, and $\mathcal{C}$ correspond to the *transformation*, *projection*, and *composition* modules.

**Transformation.** The set of 3D Gaussians is first transformed into the viewing ray coordinate system: $\mathbb{G}_t^3 = \mathcal{T}(\mathbb{G}^3)$. This step aligns the kernels with the camera rays to reduce computation overhead.

**Projection.** The transformed Gaussians are then mapped onto the image plane: $\mathbb{G}^2 = \mathcal{P}(\mathbb{G}_t^3)$. The resulting 2D Gaussians retain the opacity and color of their 3D counterparts, while the depth dimension in position and covariance is discarded.

**Composition.** Finally, an image is synthesized by alpha-compositing the 2D Gaussians in a back-to-front order: $\mathbf{I}_{rgb} = \mathcal{C}(\mathbb{G}^2)$. The differentiable rasterizer $\mathcal{R}$ enables the optimization of the kernel parameters using gradient-based losses.

## C  Implementation Details

### C.1  Dataset

We conduct experiments on the large-scale X3D [14] dataset with multiple modalities across two categories: Human Organ (Chest, Head, Foot, Jaw, Pancreas, Abdomen, aneurism, Leg) and Synthetic Object (Bonsai, Teapot, Engine, Backpack, box, carp, Pelvis). The chest scans are sourced from LIDC-IDRI [49], the pancreas scans from Pancreas-CT [59], and the remaining cases from SciVis [60]. Following prior works [15, 16, 14], we employ the TIGRE tomography toolbox [53] to capture $512 \times 512$ projections across a range of $0°$ to $180°$, including both in-distribution and out-of-distribution views.

**View Distribution.** For training, we uniformly capture 5, 10, 15, 25, and 50 views within $0°$ to $180°$. For testing, we randomly capture 50 views in the same range.

**Out-of-Distribution Views.** We also evaluate the robustness of our method on out-of-distribution views. To construct these views, we apply a consistent algorithm across all cases. Specifically, for each case, we calculate the Oriented Bounding Box (OBB) [47] of the trained ellipsoid distribution to determine its center and principal axis. Next, we select a direction $\mathbf{d}$ orthogonal to this principal axis.

Starting from the center, we move along $\mathbf{d}$ by a radius $r$ to establish the initial camera position. The camera then rotates $360°$ around the principal axis to form the view distribution. Due to differences in direction and radius, the resulting new view distribution is entirely distinct from both the training and testing sets. This setup comprehensively evaluates the model's robustness and 3D consistency under unseen views.

## C.2 Hyper-parameters

In our experiments, the position learning rate was set to an initial and final value of 0.0002, while the density and scaling learning rates were both initialized and maintained at 0.01 and 0.005, respectively. The structural dissimilarity loss weight ($\lambda_{\mathrm{DSSIM}}$) was set to 0.25. To refine the density of kernels, a minimum density threshold of $1 \times 10^{-5}$ was applied. The total number of kernels was capped at 500,000 to balance computational efficiency with accuracy.

# D Additional Quantitative Results

## D.1 Ablation on 50 views

We evaluate our method under the 50-view setting and compare it with state-of-the-art baselines. As shown in Table 6, our approach achieves the best performance in PSNR and LPIPS while maintaining SSIM comparable to X-Gaussian. These results, together with the 5- and 10-view experiments, demonstrate that our method consistently outperforms existing approaches across different input sparsity levels, from highly sparse to moderately dense scenarios.

| Method | SSIM ↑ | PSNR ↑ | LPIPS* ↓ |
|---|---|---|---|
| SAX-NeRF | 0.975 | 28.53 | 124.10 |
| X-Gaussian | 0.997 | 32.78 | 70.52 |
| R$^2$-Gaussian | 0.982 | 42.61 | 64.38 |
| **Ours** | 0.993 | 42.65 | 61.19 |

Table 6: **Novl View Synthesis with 50 input views.**

## D.2 Ablation on Computational Efficiency

We compare the computational efficiency of different approaches as shown in Table 7. Our method achieves training and memory consumption on the same order as 3DGS-based methods, while being significantly more efficient than NeRF-based baselines. In terms of inference, X-Field supports real-time rendering, offering a practical balance between quality and speed for clinical deployment.

| | NAF | SAX-NeRF | X-Gaussian | R$^2$-Gaussian | Ours |
|---|---|---|---|---|---|
| Training Time | 38m 57s | 307m 50s | 1m 55s | 3m 52s | 4m 15s |
| Inference Speed | 2.07 fps | 0.36 fps | 132 fps | 72 fps | 45 fps |
| Memory Usage | 24 GB | 27 GB | 824 MB | 968 MB | 1456 MB |

Table 7: **Computational profile comparison.** Training and inference statistics on RTX 6000 Ada GPU.

## D.3 Statistical Significance Tests for Hybrid Initialization

As shown in Table 8, all tests indicate statistically significant improvements of our hybrid initialization over both Random and FDK-based baselines. While the average gains are relatively modest, their consistency across all evaluated scenes demonstrates the reliability of the initialization strategy. This suggests that even small improvements in early-stage initialization can provide a stable advantage for subsequent optimization.

| Metric | Ours vs Random | | Ours vs FDK | |
|---|---|---|---|---|
| | $\Delta$ | $p$-val | $\Delta$ | $p$-val |
| PSNR ↑ | +0.84 | 0.0002 | +0.42 | 0.0016 |
| SSIM ↑ | +0.0071 | 0.0005 | +0.0035 | 0.0012 |
| LPIPS ↓ | −0.53 | 0.0007 | −0.29 | 0.0013 |

Table 8: **Statistical significance of hybrid initialization.**

## D.4 Component Efficiency Analysis

Table 9 reports runtime and memory for different ablation settings. Removing segment length yields the largest savings but leads to severe quality degradation, as accurate path-dependent attenuation cannot be computed. Excluding material optimization provides moderate efficiency gains but compromises material-aware reconstruction. Overlap filter removal has little effect on training time yet increases memory usage, mainly affecting efficiency. Removing intersection computation offers modest speedup with reduced memory, indicating moderate overhead. Overall, the full model introduces limited additional cost while preserving reconstruction fidelity.

| Method | $R^2$-Gaussian | w/o Material Opt. | w/o Overlap Filter | w/o Intersection | w/o Segment Length | Full |
|---|---|---|---|---|---|---|
| Training Time | 3m52s | 3m58s | 4m12s | 4m04s | 3m46s | 4m15s |
| Memory Usage | 968 MB | 1174 MB | 1359 MB | 1368 MB | 945 MB | 1456 MB |

Table 9: **Training time and memory usage under different ablations.**

## D.5 Real-World Data Validation

We evaluate our method on the FIPS dataset [61], which provides real 2D X-ray projections. As shown in Table 10, X-Field consistently outperforms baseline approaches under both 5-view and 10-view settings. Moreover, the performance trends observed on synthetic benchmarks generalize well to real projection data containing scattering, beam hardening, and detector noise, demonstrating that our approach remains robust and effective under practical real-world conditions.

| Method | 5-view | | | 10-view | | |
|---|---|---|---|---|---|---|
| | PSNR ↑ | SSIM ↑ | LPIPS ↓ | PSNR ↑ | SSIM ↑ | LPIPS ↓ |
| SAX-NeRF | 22.28 | 0.963 | 162.23 | 25.86 | 0.978 | 141.50 |
| X-Gaussian | 25.68 | 0.967 | 121.62 | 29.24 | 0.980 | 76.94 |
| $R^2$-Gaussian | 36.42 | 0.970 | 71.25 | 38.87 | 0.984 | 34.77 |
| **Ours** | 37.56 | 0.981 | 70.41 | 40.26 | 0.989 | 34.22 |

Table 10: **Performance on the FIPS real projection dataset.**

## D.6 Quantitative Comparison across Scenes

Tables 11 and 12 present the scene-wise quantitative results for 10-view and 5-view training data, respectively. In the 10-view setting (Table 11), our method achieves the best performance in most scenes across all metrics, including PSNR, SSIM, and LPIPS. Specifically, for challenging scenes such as the pancreas and bonsai, our method consistently outperforms all others, demonstrating its ability to handle complex structures effectively. The $R^2$-Gaussian method performs well in certain scenes, such as the foot and engine, achieving competitive SSIM and LPIPS scores. However, our method maintains an edge in overall quality, solidifying its superiority in reconstructing finer details.

| | Foot | | | Head | | | Chest | | | Jaw | | |
|---|---|---|---|---|---|---|---|---|---|---|---|---|
| Method | PSNR↑ | SSIM↑ | LPIPS*↓ | PSNR↑ | SSIM↑ | LPIPS*↓ | PSNR↑ | SSIM↑ | LPIPS*↓ | PSNR↑ | SSIM↑ | LPIPS*↓ |
| SAX-NeRF [14] | 17.97 | 0.943 | 213.3 | 17.25 | 0.958 | 154.6 | 15.63 | 0.937 | 187.2 | 18.07 | 0.949 | 267.9 |
| X-Gaussian [15] | 17.86 | 0.942 | 120.6 | 17.20 | 0.958 | 136.4 | 15.71 | 0.939 | 132.8 | 17.93 | 0.947 | 227.2 |
| $R^2$-Gaussian [16] | 34.46 | 0.951 | 80.73 | 34.00 | 0.972 | 86.25 | 35.46 | 0.949 | 100.4 | 30.15 | 0.909 | 175.3 |
| **Ours** | 35.54 | 0.959 | 75.56 | 35.71 | 0.975 | 76.84 | 34.03 | 0.944 | 91.15 | 29.32 | 0.910 | 164.3 |

| | Pancreas | | | Bonsai | | | Teapot | | | Engine | | |
|---|---|---|---|---|---|---|---|---|---|---|---|---|
| Method | PSNR↑ | SSIM↑ | LPIPS*↓ | PSNR↑ | SSIM↑ | LPIPS*↓ | PSNR↑ | SSIM↑ | LPIPS*↓ | PSNR↑ | SSIM↑ | LPIPS*↓ |
| SAX-NeRF [14] | 18.78 | 0.929 | 168.5 | 21.28 | 0.971 | 174.5 | 23.67 | 0.985 | 122.8 | 21.89 | 0.977 | 127.2 |
| X-Gaussian [15] | 18.77 | 0.931 | 121.7 | 21.66 | 0.977 | 91.83 | 24.24 | 0.986 | 66.71 | 22.14 | 0.977 | 72.29 |
| $R^2$-Gaussian [16] | 36.39 | 0.968 | 79.36 | 42.12 | 0.983 | 46.85 | 40.37 | 0.980 | 34.03 | 39.82 | 0.990 | 23.44 |
| **Ours** | 36.77 | 0.969 | 81.25 | 43.06 | 0.987 | 45.52 | 38.03 | 0.988 | 36.97 | 41.79 | 0.993 | 20.18 |

Table 11: **Results of Quantitative Comparison on 10 views (§ D.6).** We compare our X-Field on 10 views with Deep Learning-based methods: SAX-NeRF [14], X-Gaussian [15], and $R^2$-Gaussian [16]. LPIPS* = LPIPS $\times 10^3$. best and second best methods are marked.

In the 5-view scenario (Table 12), the performance gap between methods decreases due to the limited training data. Nonetheless, our method achieves the best results in most scenes, particularly excelling in the chest, bonsai, and teapot, where it surpasses other methods across all metrics. While the $R^2$-Gaussian method demonstrates strong performance in specific scenes, our approach consistently proves to be more robust. These results underscore our method's adaptability to reduced training data while maintaining state-of-the-art performance.

| Method | Foot PSNR↑ | SSIM↑ | LPIPS*↓ | Head PSNR↑ | SSIM↑ | LPIPS*↓ | Chest PSNR↑ | SSIM↑ | LPIPS*↓ | Jaw PSNR↑ | SSIM↑ | LPIPS*↓ |
|---|---|---|---|---|---|---|---|---|---|---|---|---|
| SAX-NeRF [14] | 18.21 | 0.922 | 234.7 | 15.23 | 0.927 | 192.2 | 13.67 | 0.813 | 237.3 | 16.16 | 0.917 | 322.4 |
| X-Gaussian [15] | 18.48 | 0.925 | 147.8 | 15.19 | 0.929 | 177.2 | 13.61 | 0.824 | 201.1 | 16.13 | 0.918 | 306.7 |
| R²-Gaussian [16] | 30.32 | 0.946 | 117.2 | 31.25 | 0.958 | 108.6 | 27.65 | 0.862 | 169.1 | 26.43 | 0.850 | 232.9 |
| **Ours** | 30.88 | 0.951 | 116.2 | 31.85 | 0.960 | 97.53 | 29.73 | 0.899 | 146.4 | 25.16 | 0.833 | 227.8 |

| Method | Pancreas PSNR↑ | SSIM↑ | LPIPS*↓ | Bonsai PSNR↑ | SSIM↑ | LPIPS*↓ | Teapot PSNR↑ | SSIM↑ | LPIPS*↓ | Engine PSNR↑ | SSIM↑ | LPIPS*↓ |
|---|---|---|---|---|---|---|---|---|---|---|---|---|
| SAX-NeRF [14] | 18.57 | 0.924 | 188.3 | 19.25 | 0.953 | 209.6 | 21.46 | 0.975 | 160.9 | 21.87 | 0.960 | 116.2 |
| X-Gaussian [15] | 18.60 | 0.927 | 145.5 | 19.43 | 0.958 | 148.2 | 22.03 | 0.978 | 121.8 | 22.25 | 0.965 | 94.87 |
| R²-Gaussian [16] | 30.84 | 0.937 | 103.4 | 34.85 | 0.966 | 83.27 | 27.69 | 0.945 | 77.56 | 31.73 | 0.968 | 52.92 |
| **Ours** | 30.99 | 0.944 | 100.2 | 34.73 | 0.965 | 82.64 | 28.30 | 0.950 | 76.45 | 32.13 | 0.970 | 52.13 |

Table 12: **Results of Quantitative Comparison on 5 views (§ D.6).** We compare our X-Field on 5 views with Deep Learning-based methods: SAX-NeRF [14], X-Gaussian [15], and R²-Gaussian [16]. LPIPS* = LPIPS $\times 10^3$. best and second best methods are marked.

# E  Additional Qualitative Results

**Out-of-Distribution Results**  To further evaluate X-ray reconstruction capabilities, we rendered 360-degree images around the scene. We compared X-Gaussian, R²-Gaussian , and our method with viewpoints being out-of-distribution in 5 different scenes, including foot (Figure 9), head (Figure 10), jaw (Figure 11), and teapot (Figure **??**) .

Each figure presents five out-of-distribution views, with rows representing methods and columns corresponding to the same view. In the foot scene (Figure 9), X-Gaussian exhibits black linear artifacts in views 3 and 4. While R²-Gaussian reconstructs the general foot structure, our method delivers smoother and more visually consistent results. In the head scene (Figure 10), X-Gaussian displays wave-like artifacts at the top of the head, whereas both R²-Gaussian and our method achieve improved overall reconstructions. Notably, our method captures finer details, showcasing superior reconstruction quality and generalization. Similarly, in the jaw (Figure 11)) ) scenes, our method produces clearer and more complete reconstructions compared to the other approaches.

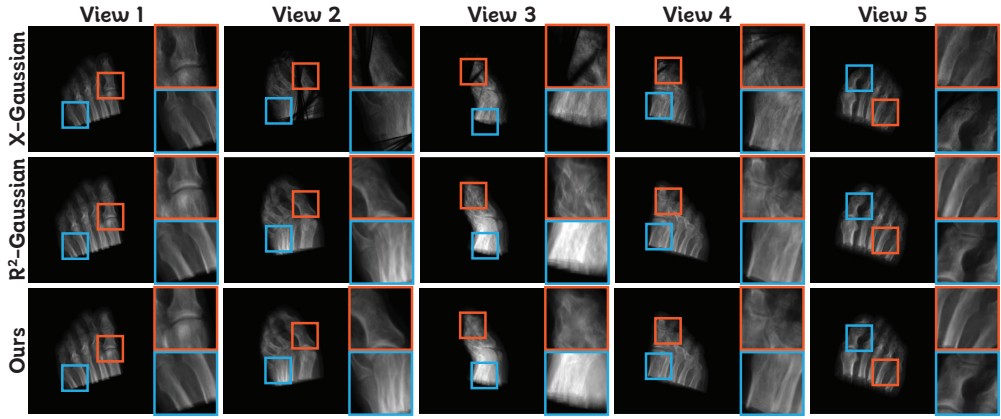

Figure 9: **Qualitative Comparison of Foot on Out-of-Distribution Views.**

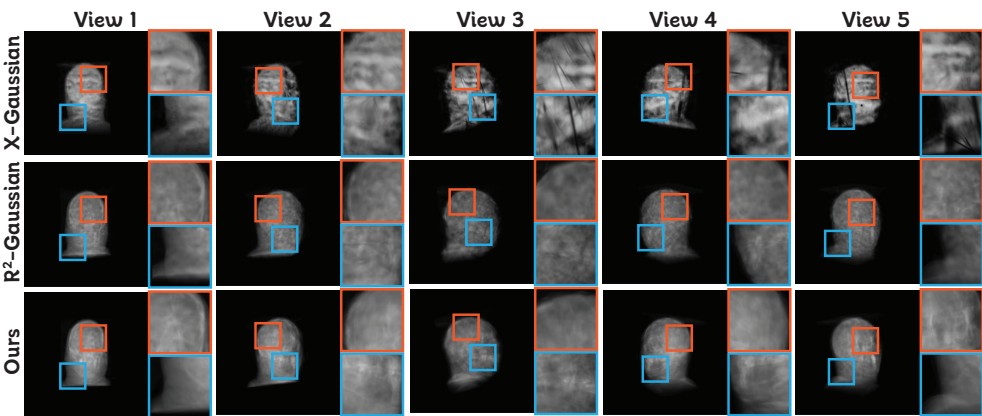

Figure 10: **Qualitative Comparison of Head on Out-of-Distribution Views.**

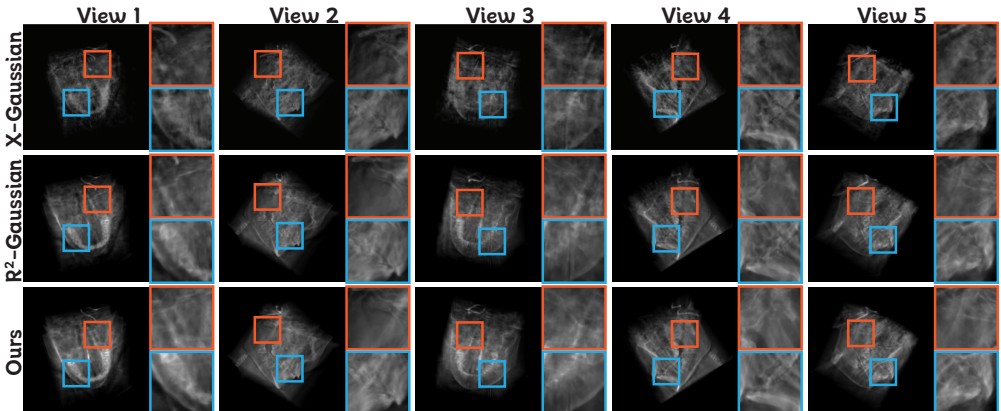

Figure 11: **Qualitative Comparison of Jaw on Out-of-Distribution Views.**

# F More Discussions

## F.1 Limitations

Despite the significant advancement achieved by X-Field in X-ray sparse-view reconstruction, it is not without limitations:

- **The ellipsoid representation may not be the most efficient.** We use ellipsoids to capture the internal material distribution of objects; however, there could exist a more efficient form that can reduce the number of parameters while also requiring fewer elements to effectively learn the material distribution.

- **Lack of leveraging large model prior knowledge.** Many recent methods for sparse view reconstruction have utilized prior knowledge from large models to guide their learning processes. In our task, we didn't find a suitable large model for X-ray data to introduce into our framework. It is possible that some large-scale medical models could provide prior knowledge to enhance convergence speed and improve overall performance.

## F.2 Ethics Considerations

Ethical considerations are critical in developing X-ray reconstruction technologies. Ensuring informed consent for transparent communication about and limitations of X-ray reconstruction methods is essential to respect privacy and prevent misrepresentation. Secure handling and storage of medical imaging data are paramount to safeguard against unauthorized access and misuse. Furthermore, recognizing the potential risks associated with the misuse of advanced imaging technologies, we emphasize the importance of establishing robust ethical guidelines to ensure their responsible application. Our commitment is to uphold the highest ethical standards in all aspects of X-ray reconstruction, protecting the integrity and confidentiality of patient data.

