# OpenReview forum: "X-Field: A Physically Informed Representation for 3D X-ray Reconstruction"
_NeurIPS.cc/2025/Conference — NeurIPS 2025 spotlight_

### Official Review · Reviewer_Xpaz · 2025-06-21

**Clarity:** 3
**Significance:** 3
**Originality:** 3
**Rating:** 5
**Confidence:** 3

**Summary:**

This paper proposes a 3DGS variation adapted for 3D X-rays, by representing an object's structure using ellipsoids with distinct attenuation coefficients. This strategy models the radiological density across various materials and accounts for penetration properties, unlike previous 3DGS approaches like $\text{R}^2$-Gaussian, which achieves maximum density at the ellipsoid center and gradually decays outwards. Additionally, the authors introduce an intersection algorithm to handle overlapping ellipsoids and employ hybrid initialization and material-based optimization strategies. The experiments on real and synthetic datasets demonstrate that X-Field outperforms current methods in 3D CT reconstruction and X-ray novel view synthesis.

**Questions:**

- Could the authors clarify the computational runtime and memory for each component, and also with respect to approaches such as $\text{R}^2$-Gaussian.
- How robust is the method to initialization, e.g. if the ellipsoid positions are substantially off? And how robust is the method to different hyperparameters?
- What cases does the method struggle with, e.g. how does it perform in vascular networks or heterogeneous tissues? Clarifying potential failures would be beneficial.

**Ethical Concerns:**

["NO or VERY MINOR ethics concerns only"]

**Final Justification:**

The authors have addressed my concerns during the rebuttal, conducting new experiments that demonstrate reasonable computational requirements and robustness in different scenarios.

**Limitations:**

Yes

**Paper Formatting Concerns:**

No major formatting issues were identified.

**Quality:**

3

**Strengths And Weaknesses:**

The main strengths are:

- The paper is well-written, easy to follow, with clear handling of ellipsoid intersections.
- The work is clearly motivated by the physics of X-ray imaging, addressing limitations of previous visible-light-based representations.
- The paper has appropriate experiments, showing consistent improvements across multiple challenging datasets, both quantitatively and qualitatively.

The main weaknesses are:

- The results only offer a small visual improvement over $\text{R}^2$-Gaussian, yet the paper doesn't sufficiently detail performance/memory requirements, making it difficult to assess how beneficial this work will be to the community. However, the supplementary material video helps convince of the quality improvement.
- The new parameterization and pipeline stages seem slightly more involved (and potentially slower) than for example $\text{R}^2$-Gaussian. While an ablation study is included, it's not clear if there are elements here that are dependent on the full new pipeline or can be used in isolation with concurrent 3DGS strategies, and how they contribute to the performance timings/memory. Similarly, as 3DGS benefits from more iterations/elements, it's not clear how hyperparameter tuning influences the results, given the small visual improvements.

---

> ### Author Rebuttal · Authors · 2025-07-30
>
> We thank the reviewer for detailed review and constructive suggestions, which are helpful in improving our paper.
> # Weakness 1.1: Visual improvement
> We acknowledge that the visual improvements in Figure 6 appear modest compared to R²-Gaussian. However, as you noted, the supplementary video demonstrates more significant quality differences, particularly in terms of smoother reconstructions and reduced artifacts. The varying visual improvement magnitudes may be attributed to different evaluation conditions: Figure 6 results are obtained from test distribution views, while the supplementary video showcases out-of-distribution (OOD) views, which present greater reconstruction challenges and thus highlight our method's advantages more clearly.
>
> The supplementary material video showcases our method's performance on out-of-distribution (OOD) views. As detailed in Section C.1 and visualized in Supplementary Material E, we evaluate 360-degree novel view synthesis where cameras are positioned orthogonally to the principal axis at varying radii, creating viewpoints entirely distinct from the training distribution.
>
> Under these challenging OOD conditions, our method demonstrates superior generalization capabilities and enhanced robustness to unseen viewpoints. This stems from our physically grounded representation that better captures X-ray physics, enabling consistent performance across diverse viewing scenarios.
>
>
> The supplementary material video showcases our method's performance on out-of-distribution (OOD) views. As detailed in Section C.1 and visualized in Supplementary Material E, we evaluate 360-degree novel view synthesis where cameras are positioned orthogonally to the principal axis at varying radii, creating viewpoints entirely distinct from the training distribution.
>
> Under these challenging OOD conditions, our method demonstrates superior generalization capabilities and enhanced robustness to unseen viewpoints. This stems from our physically grounded representation that better captures X-ray physics, enabling consistent performance across diverse viewing scenarios.
>
>
>
> # Weakness 1.2: Computational requirements
> We provide detailed computational analysis for reconstructing 10 input views with 5,000 iterations on an RTX 6000 Ada GPU:
>
> |                | NAF        | SAX-NeRF   | X-Gaussian | R²-Gaussian | Ours      |
> |----------------|------------|------------|------------|-------------|-----------|
> | Training Time  | 38 min 57s| 307 min 50s | 1min 55s | 3 min 52s  | 4min 15s |
> | Inference Speed |  2.07 fps | 0.36 fps | 132 fps | 72 fps | 45 fps |
> | Memory Usage   | 24GB       | 27GB       | 824MB      | 968 MB      | 1456 MB   |
>
> Our method significantly outperforms NeRF-based approaches across all metrics: substantially faster training, considerably faster inference, and significantly lower memory usage.  Compared to 3DGS methods, we achieve comparable computational efficiency while delivering superior reconstruction quality. We appreciate the reviewer's valuable feedback and will incorporate the computational analysis in the final version.
>
>
> # Weakness 2: Component Dependencies
>
> Our method introduces a physically-grounded representation using homogeneous 3D ellipsoids with distinct attenuation coefficients to model diverse materials. This foundational representation requires
>
>  **Tightly-coupled specialized components:**
>
> - Segment length calculation: Computes X-ray path lengths through ellipsoidal volumes
>
> - Intersection correction: Handles overlapping ellipsoid scenarios
>
> - Material-based optimization: Refine ellipsoid distribution along material boundaries.
>
> These components are intrinsically interdependent as they collectively implement our physics-based ellipsoidal representation. Ablation results in the Table 3 confirm this coupling - removing segment length calculation drops SSIM from 35.03 to 27.48.
>
>
> **Modular Components:**
> While the core representation requires full pipeline integration, our Faithfully Overlap Filter is modular and can be independently applied to other 3DGS-based X-ray reconstruction methods.
>
> **Performance Timings/Memory:** We provide detailed analysis of computational runtime and memory usage for each component in **Question 1** below.
>
> **Hyperparameter:** For all training processes, we trained all methods using the same number of iterations (20,000) to ensure fair comparison. We provide more comprehensive analysis on hyperparameter sensitivity in **Question 2.2** below.
>
>
>
>
>
> # Question 1: Computational runtime and memory for each component
>
>
> We appreciate this important question and provide detailed computational analysis for each component in the table below.
>
> | Method | R²-Gaussian | w/o Material Opt. | w/o Overlap Filter | w/o Intersection | w/o Segment Length | Full |
> |---------|-------------|-------------------|-------------------|------------------|-------------------|------|
> | Training Time | 3 min 52s | 3min 58s | 4min 12s | 4min 04s | 3min 46s | 4 min 15s |
> | Memory Usage | 968 MB | 1174 MB | 1359 MB | 1368 MB | 945 MB | 1456 MB |
>
> Component Analysis: Removing segment length calculation results in the most significant computational savings in both training time and memory usage. However, this severely degrades rendering quality as the model cannot accurately compute X-ray attenuation along ray paths, which is fundamental to our method. Material optimization removal provides moderate computational benefits but compromises material-aware reconstruction capabilities. Overlap filter elimination shows minimal training time impact while slightly increasing memory usage, indicating this component primarily affects memory efficiency. Intersection calculation removal offers modest speedup with reduced memory consumption, demonstrating its moderate computational overhead. We will include this computational analysis in the final version of the paper.
>
>
>
>
> # Question 2.1: Initialization Robustness
>
> Our method demonstrates robust performance across different initialization strategies. As illustrated in Figure 4, we evaluate three initialization approaches: (1) random Gaussian initialization of ellipsoids (producing scattered, uninformative starting positions), (2) FDK-based initialization providing coarse geometric structure, and (3) our hybrid initialization with detailed geometry information.
>
> As shown in Table 4, even with random initialization representing substantially poor starting positions (Figure 4, left), our method achieves competitive performance, demonstrating strong convergence capability from uninformative initial states. Progressive improvement is observed with better initialization: FDK initialization achieves 37.96, while our hybrid approach reaches optimal performance.
>
> This robustness stems from our physics-based optimization framework, which guides ellipsoids toward correct positions through X-ray consistency constraints regardless of starting positions. However, more informative geometry initialization further accelerates convergence and helps the model achieve superior final performance by providing better starting points closer to optimal solutions.
>
>
>
> # Question 2.2: Robustness to hyperparameters
>
>
> To evaluate hyperparameter robustness, we conducted two experiments on human organ data using 10 input views, analyzing the sensitivity to training iterations and learning rates across different hyperparameter configurations.
>
> **Training Iteration Analysis:**
>
> Our first experiment examines performance across different training iterations:
>
> | Iteration | 5000  | 10000 | 20000 |
> |---------|-------|-------|-------|
> | PSNR    | 34.78 | 35.18 | 35.71 |
> | SSIM    | 0.960 | 0.943 | 0.980 |
> | LPIPS   | 73.26  | 72.14  | 71.03 |
>
> The results demonstrate stable performance across different iteration counts, with all configurations achieving competitive quality metrics.
>
> **Learning Rate Sensitivity:**
>
> Our second experiment evaluates robustness to learning rate variations:
>
> | Learning Rate | PSNR | SSIM | LPIPS |
> |---------------|---------|---------|-------|
> | × 0.5         | 33.540  | 0.962   | 71.59 |
> | × 1 (default) | 34.801  | 0.971   | 71.26 |
> | × 2           | 34.767  | 0.966   | 71.48 |
>
> The results show consistent performance across different learning rates, with only modest variations in quality metrics. Both halved and doubled learning rates maintain competitive reconstruction quality compared to the default setting.
>
> **Conclusion:** These experiments demonstrate that our method exhibits strong robustness to key hyperparameters, maintaining stable performance across different training iterations and learning rate configurations.
>
>
> # Question 3: Cases method struggle with.
>
> For vascular networks, the low inherent contrast in conventional X-ray images poses challenges for our reconstruction method, as fine vascular structures may not be adequately captured. However, given contrast-enhanced imaging, the underlying formulation of our method is readily compatible, offering a promising solution.
>
> For heterogeneous tissues, as demonstrated in our foot (Figure 6) and head (Figure 7) examples, our method successfully differentiates between distinct tissue types such as bone and muscle. However, achieving cellular-level precision in highly heterogeneous regions remains an area for future exploration.

---

> > ### Comment · Reviewer_Xpaz · 2025-08-05
> >
> > Thank you for the response and additional experiments, which have addressed my concerns. I have raised my rating to accept (score = 5). If possible, please add an example (success or fail cases) from vascular networks, perhaps contrast enhanced MR angiography or similar in the final paper or SM, so the community can have realistic expectations as to how it performs in this setting.

---

> > > ### Author Response · Authors · 2025-08-06
> > >
> > > We sincerely thank the reviewer for the thoughtful comments and are glad to hear that the concerns have been addressed. We also appreciate the constructive suggestion regarding vascular networks. We are currently identifying suitable contrast-enhanced angiography data and intend to include preliminary reconstruction results in the supplementary material as part of our ongoing extension. We agree this would be a meaningful addition and could help clarify the method's applicability in complex vascular scenarios.

---

### Official Review · Reviewer_u2wa · 2025-06-30

**Clarity:** 4
**Significance:** 4
**Originality:** 4
**Rating:** 5
**Confidence:** 3

**Summary:**

This paper presents X-Field, a novel physically grounded 3D representation for X-ray imaging. The key contributions include the use of homogeneous 3D ellipsoids with distinct attenuation coefficients to model diverse materials within internal structures, an efficient path partitioning algorithm to resolve the intersection of ellipsoids and compute cumulative attenuation along an X-ray path, a hybrid progressive initialization to refine the geometric accuracy, and a material-based optimization to enhance model fitting along material boundaries.

**Questions:**

Please see the Weaknesses 1 2 3
1. Clarification of OBB and X-Field's Physically Faithful Overlap Filtering Differences.
2. Qualitative Results for Ablation Studies.
3. Consideration of Polychromatic X-ray Sources: Would the  X-Field results in artifact when face the polychromatic X-ray sources?

**Ethical Concerns:**

["NO or VERY MINOR ethics concerns only"]

**Final Justification:**

This paper present a novel design for 3DGS in CT novel view sythnsis.
Considering the views of other reviewers that this work seems to combine NeRF's ray tracing integration with 3DGS, I think this is an innovative combination. Because unlike some works that combine NeRF and GS that simply and directly introduce the dual-stream rasterization and ray tracing process, X-Field ingeniously makes up for the shortcomings of 3DGS by integrating only within the ellipsoid. At the same time, it does not have the drawback of slow training introduced by NeRF.
My concern of Weakness 1,2,3 has been solved by authors response.
Therefore my score is accept(5)

**Limitations:**

yes

**Quality:**

4

**Strengths And Weaknesses:**

### **Strengths**:

1. The uses of material-adaptive ellipsoids with distinct attenuation coefficients, is a novel and physically meaningful approach tailored for X-ray and CT.
2. The proposed segment length calculation with intersections and physically faithful overlap filtering help calculate pixel values more accurately.
3. The hybrid progressive initialization helps for efficient optimization and better convergence.
3. The method demonstrates superior performance in both X-ray NVS and CT reconstruction tasks across various datasets compared to state-of-the-art methods, especially under sparse input view.
### **Weakness**
1. Clarification of OBB and X-Field Differences during the splatting process:
How to "filter out the irrelevant pixels while preserving only the valid tiles" after OBB process.
2. Lacking qualitative results for ablation studies, which would offer more comprehensive insights into the impact of each component on the overall performance and visual quality of the reconstructions.
3. The method does not account for the polychromatic nature of X-ray sources, which can lead to beam hardening artifacts.

---

> ### Author Rebuttal · Authors · 2025-07-30
>
> Thank you very much for your acknowledgement of our work and your constructive feedbacks and suggestions.
>
> # Weakness 1 & Question 1: OBB vs. Physically Faithful Overlap Filtering
> **Our process after OBB:**
> Building upon the eigenvalue decomposition performed during OBB computation, our X-Field method utilizes the extracted principal axis lengths (eigenvalues $\lambda_1$, $\lambda_2$) and rotation angle $\theta$ from the 2D covariance matrix. We compute directional projection extents using the formula:
>
> $$\text{proj}_{x} = \max(|\lambda_1 \cos(\theta)|, |\lambda_2 \sin(\theta)|)$$
>
> $$\text{proj}_{y} = \max(|\lambda_1 \sin(\theta)|, |\lambda_2 \cos(\theta)|)$$
>
> This determines the maximum extents in horizontal and vertical directions, enabling us to establish tight-fitting rectangular boundaries that conform to the ellipse's actual shape. We then precisely identify relevant pixel regions while filtering out irrelevant areas by iterating only through tiles within the computed rectangular bounds, thereby directly excluding tiles that fall outside the ellipse's effective coverage without requiring explicit per-pixel testing.
>
> **Difference from OBB:** Unlike OBB which uses axis-aligned rectangular bounds ignoring ellipse orientation, our method incorporates the computed ellipse orientation and principal axes information. This determines orientation-aware bounds that more accurately capture the ellipse's spatial footprint, achieving tighter boundaries and reducing unnecessary tile processing.
>
>
>
>
>
>
>
>
>
> # Weakness 2 & Question 2: Qualitative Results for Ablation Studies
>
> Thanks for your suggestion.
>
> Our current manuscript includes qualitative comparisons across different input views (5, 10, and 15 views), showing consistent performance enhancement with increased input data. We plan to conduct the following qualitative comparison experiments:
>
> **Component Ablation Visualization** (corresponding to Table 3):
>
> (1) w/o Material-Based Optimization
> (2) w/o Physically Faithful Overlap Filter (using AABB instead of our approach)
> (3) w/o Intersection Module
> (4) w/o Segment Length
> (5) Full method
>
> **Initialization Method Visualization** (corresponding to Table 4):
>
> (1) Random initialization
> (2) FDK initialization
> (3) Our hybrid initialization
>
> We will include these additional qualitative ablation results in the camera-ready version.
>
>
>
>
>
> # Weakness 3 & Question 3: Polychromatic X-ray Sources
>
> Thanks for raising the important point about polychromatic X-ray sources. Our method, following current works on ray-level modeling (SAX-NeRF, X-Gaussian, R2-Gaussian), focuses on monochromatic X-ray reconstruction rather than polychromatic X-ray sources. The X3D dataset we used also adopts monochromatic assumptions, generated using the TIGRE tomography toolbox with single-energy X-ray simulations.
>
> To better model polychromatic X-ray sources, photon-level modeling of X-ray imaging physics would be a promising direction to capture energy-dependent attenuation effects. This represents an excellent future research direction that we plan to explore in our subsequent work. Thanks for your insightful suggestion.

---

> > ### Comment · Reviewer_u2wa · 2025-08-01
> > **Thanks for response**
> >
> > My concern of Weakness 1 has been solved.
> > Wishing to see the visualization added in the final version(Weakness2).
> > Besides, it is understandable to follow the prerequisite work and only consider monochromatic X-rays(Weakness3).
> > Thanks for author's response.

---

> > > ### Author Response · Authors · 2025-08-02
> > >
> > > Thank you for your valuable feedback and the time you took to review our paper. We are glad that our response addressed your concerns. We sincerely appreciate your suggestions and will incorporate additional visualization in the final version of our paper.

---

### Official Review · Reviewer_VmXW · 2025-07-02

**Clarity:** 1
**Significance:** 2
**Originality:** 3
**Rating:** 5
**Confidence:** 4

**Summary:**

This work proposes X-Field, a 3D Gaussian Splatting (3D GS)-based method for reconstructing 3D CT volumes from sparse-view X-ray projections. Technically, the main contributions of this work are as follows: 1) proposing a material-adaptive ellipsoids representation for improved accuracy; 2) building a hybrid progressive initialization pipeline for enhancing initial point clouds of 3D GS; 3) using a material-based optimization to improve compact and efficient representations. On X3D, a large-scale simulated CT dataset, the empirical results show that the proposed X-Field produces state-of-the-art (SOTA) performance in both X-ray novel view synthesis and CT reconstruction.

**Questions:**

See Strengths and Weaknesses above, please.

**Ethical Concerns:**

["NO or VERY MINOR ethics concerns only"]

**Final Justification:**

During the rebuttal, the authors have addressed my concerns about "material adaptive" and evaluation on real-world data well. Therefore, I have raised my rating to accept.

**Limitations:**

yes

**Quality:**

3

**Strengths And Weaknesses:**

### Strengths
+ In the proposed material-adaptive ellipsoids, the algorithm to partition rays into segments intersecting overlapping ellipsoids is elegant and practical. It can ensure correct X-ray integration and eliminate complex overlaps.
+ The proposed hybrid initialization is well-engineered and reasonable.
+ On a simulated dataset, the proposed X-Field performs better compared to the latest 3D GS-based methods, such as X-Gaussian and R^2-Gaussian.
+ This work conducts comprehensive ablation studies, confirming the effectiveness of several components of the proposed X-Field.

---

### Weaknesses

+ This work claims that the proposed X-Field is a physically grounded representation. However, it merely converts the continuous integral transform (Eq. 2) into a discrete summation (Eq. 3), which is a common practice in previous NeRF-based CT reconstruction methods (e.g., NAF and SAX-NeRF). Moreover, the term “material-adaptive” is very vague. Specifically, by using the segment length, any location in the imaging space is dominated by a single ellipsoid. But why does this make it “material-adaptive”? The term “material” is commonly used in quantitative CT, such as in material decomposition. Here, does “material-adaptive” refer to material decomposition?
+ The hybrid initialization is fully empirical and involves multiple pre-processing methods. From the results in Table 4, its effectiveness seems to be very minor.
+ All the experiments are conducted on a single simulated dataset. As a new reconstruction technique for medical imaging, I think it is necessary to test its effectiveness on clinical data.
+ The results do not report the computational costs (running time and memory footprint) of the proposed X-Field. As a 3D CT reconstruction technique, its computational efficiency is critical for clinical use.
+ The writing could be improved for clarity. In particular, readers who are not familiar with 3D GS may find the paper hard to follow.

---

> ### Author Rebuttal · Authors · 2025-07-30
>
> We thank the reviewer for the detailed review. The comments and suggestions are helpful in improving our paper.
>  # Weakness 1.1: Physically Grounded Representation
>
> We acknowledge that discrete summation is common across methods, but our contribution lies in the representation design and rendering approach specifically tailored for X-ray imaging physics.
>
> **Different Origins and Motivations:**
>
> - NeRF-based methods (NAF, SAX-NeRF):
>
> These originate from NeRF, which is designed for volumetric rendering. They implicitly encode information with networks, predicting radiodensity along rays and accumulating for rendering, rather than explicitly representing material distribution.
>
> - 3DGS-based methods (X-Gaussian, R²-Gaussian):
>
> They explicitly model with Gaussian ellipsoids, where density peaks at ellipsoid centers and decreases toward edges. They render images by projecting 3D Gaussian ellipsoids into 2D planes in a splatting manner, instead of considering physically accurate ray traversal through 3D space.
>
> - Our approach:
>
> Our work originates from X-ray imaging principles and extends 3DGS for physically grounded X-ray rendering. We use homogeneous 3D ellipsoids with distinct attenuation coefficients representing different materials. Following Beer-Lambert law, we explicitly calculate the path length X-rays traverse through each ellipsoid and compute cumulative attenuation along the entire X-ray path for physically accurate rendering. During optimization, ellipsoids are updated based on X-ray physics constraints to ensure the representation matches the actual X-ray attenuation process.
>
> **Key Physical Modeling Differences:** Our method explicitly implements physically meaningful parameters with homogeneous attenuation coefficients and precise path lengths, rather than generic volumetric rendering or splatting operations. While the mathematical form appears similar (discrete summation), the underlying physical modeling, representation design, and parameter optimization are fundamentally different and specifically grounded in X-ray imaging principles.
>
>
> # Weakness 1.2:  "Material-Adaptive" Definition
> From a high-level perspective, "material-adaptive" refers to our method's ability to automatically adjust ellipsoid representation to capture different material distributions within the X-ray imaging space. Unlike traditional material decomposition in quantitative CT (which separates known materials like bone and tissue), our approach adaptively refines the ellipsoid placement and attenuation coefficients to model diverse internal materials without prior knowledge of their specific types.
>
> The "adaptive" aspect is achieved through our use of homogeneous 3D ellipsoids with distinct attenuation coefficients, positions, and covariances. During optimization, the model updates ellipsoids’ properties, and adaptively clones and splits ellipsoids to model diverse materials within internal structures, ensuring the representation matches the actual material distribution.
>
> This differs from material decomposition in that we do not decompose into predefined material types, but rather adaptively discover and model the material distribution patterns present in the data through our ellipsoid-based representation and optimization process.
>
> # Weakness 2: Hybrid Initialization
> Our hybrid approach follows a principled coarse-to-fine strategy: CGLS provides global structural estimation, SART refines local details, and TV regularization preserves edges while suppressing noise. Each component serves a specific purpose based on established CT reconstruction principles rather than empirical selection.
>
> Our method requires comparable computational time to FDK but achieves better performance and provides improved geometric priors for the subsequent optimization process.
>
> # Weakness 3: Effectiveness on Clinical Data.
>
> Following previous X-ray reconstruction works (SAX-NeRF, X-Gaussian, R²-Gaussian), we use the X3D dataset [1], which includes substantial real clinical data evaluation rather than being limited to simulated data. The X3D dataset contains two collections: Human Organs derived from real clinical medical datasets, and Daily Objects from synthetic datasets, with projections captured using the TIGRE tomography toolbox with realistic scatter and noise modeling.
>
> **Real Clinical Data Sources in X3D:**
>
> - Chest scans from LIDC-IDRI [2]: A widely-used clinical lung CT database containing real patient data
>
> - Pancreas scans from Pancreas-CT [3]: Clinical pancreatic CT data from real medical cases
>
> - Additional human organ data from VOLVIS [4]: Clinical foot, head, and jaw CT scans
>
> **Clinical Data Performance:** Our results on real clinical data are reported in Tables 1, 6, and 7, with detailed scene-wise breakdowns in supplementary materials. The human organ results demonstrate our method's superior performance across most metrics for both 5-view and 10-view inputs, confirming effectiveness on clinical data.
>
>
>
>
> # Weakness 4: Computational Costs
>
> We provide comprehensive computational cost analysis comparing our method with state-of-the-art approaches for reconstructing 10 input views with 5,000 iterations on an RTX 6000 Ada GPU, including two NeRF-based methods (NAF, SAX-NeRF) and two 3DGS-based methods (X-Gaussian, R²-Gaussian):
>
> |                | NAF        | SAX-NeRF   | X-Gaussian | R²-Gaussian | Ours      |
> |----------------|------------|------------|------------|-------------|-----------|
> | Training Time  | 38 min 57s| 307 min 50s | 1min 55s | 3 min 52s  | 4min 15s |
> | Inference Speed |  2.07 fps | 0.36 fps | 132 fps | 72 fps | 45 fps |
> | Memory Usage   | 24GB       | 27GB       | 824MB      | 968 MB      | 1456 MB   |
>
>
> **Clinical Efficiency:** Our method achieves real-time inference, delivering substantial speedup over NeRF-based methods while requiring significantly less memory. Compared to 3DGS methods, we maintain comparable training efficiency and memory usage while providing superior reconstruction quality (Table 1).
>
> **Clinical Deployment:** The combination of fast training, real-time inference, and moderate memory requirements makes our approach well-suited for clinical workflows where both reconstruction accuracy and computational efficiency are essential.
>
> *We thank the reviewer for this important suggestion and will include this computational analysis in the final version.*
>
>
>
> # Weakness 5:  3DGS Preliminary
>
> Thank you for this suggestion. We will add detailed 3DGS background material in the supplementary materials to improve clarity.
>
>
>
> [1] Structure-Aware Sparse-View X-ray 3D Reconstruction. CVPR 24.
>
> [2] The lung image database consortium (lidc) and image database resource initiative (idri): a completed reference database of lung nodules on ct scans. Medical physics 2011.
>
> [3] Data from pancreas-CT. The Cancer Imaging Archive 2016.
>
> [4] Teem nrrd volume visualization datasets. 2022.

---

> > ### Comment · Reviewer_VmXW · 2025-08-02
> >
> > Thanks for your responses. However, several of my major concerns remain unaddressed.
> >
> > 1.	First, the statement that “Our work originates from X-ray imaging principles” is an overclaim. Both existing 3D GS-based and NeRF-based methods for CT reconstruction are also grounded in the Beer–Lambert law. The proposed X-Field likewise follows this law. Moreover, precise path lengths can also be handled using non-uniform sampling strategies, such as hierarchical sampling in raw NeRF. In summary, from a modeling perspective, this work does not incorporate more X-ray physics than existing methods.
> >
> > 2.	Second, the explanation of “material-adaptive” is unconvincing. Why should adjusting the ellipsoid representation enable capturing different material distributions? In fact, material decomposition in CT is a challenging problem that typically requires dual-energy or photon-counting CT scanners, along with energy-dependent forward models. This work only uses a simple energy-independent integral model and single-energy projections. Why, then, should it be capable of reconstructing material distributions?
> >
> > 3.	In Table 4, the improvements from hybrid initialization are minor (<1 dB in PSNR, <0.01 in SSIM). Are any statistical significance tests (e.g., t-tests) reported?
> >
> >
> > 4.	The clinical data I refer to are raw projections, not simulated projections generated from medical volumes using synthetic CT geometry.

---

> ### Author Response · Authors · 2025-08-03
>
> # 1. Clarification on Physical Modeling Basis
>
> Thank you for the follow-up. We agree that the Beer–Lambert law forms the basis for many CT reconstruction methods, including NeRF-based (e.g., NAF, SAX-NeRF) and 3DGS-based approaches. Our method also follows this model, and we will revise the manuscript to avoid overstating its novelty in this regard.
>
> Our use of “X-ray imaging principles” was meant to highlight the explicit modeling of attenuation and geometry using ellipsoids—each with defined position, shape, and attenuation—rather than suggest a new physical basis. Unlike NeRF-based methods that learn radiodensity implicitly, our representation enables:
>
> - analytical ray-ellipsoid intersections,
> - segment-wise attenuation modeling,
> - resolution of overlaps for consistency.
>
> While NeRF methods can use non-uniform sampling for better integration, these operate over implicit fields. In contrast, our model uses an explicit structure more directly aligned with physical ray paths. We will revise our wording to clarify that our approach follows standard physics but emphasizes interpretability and structure.
>
>
> # 2. Clarification on the Term “Material-Adaptive”
>
> Thank you for raising this important point. As mentioned in our initial response, our method does **not** perform material decomposition in the conventional sense, which typically requires dual-energy or photon-counting CT and energy-dependent models.  Our framework is based on a standard single-energy integral model and does not attempt to distinguish or classify specific material types such as bone or soft tissue.
>
> Our goal is to enable high-fidelity novel view synthesis and improve sparse-view CT through accurate projection modeling. We optimize ellipsoid parameters (shape, location, attenuation) to match observed projections, allowing the model to represent spatially varying attenuation without inferring  specific material types.
>
> The term “material-adaptive” was originally intended to reflect this behavior. However, we acknowledge that it could imply a scope closer to quantitative CT or decomposition. To avoid confusion, we propose replacing it with **“attenuation-adaptive”** throughout the manuscript to reflect our method’s technical focus on modeling continuous, spatially-varying attenuation.  We welcome the reviewer’s feedback on whether this adjustment more clearly conveys the intended contribution and scope.
>
> # 3. Statistical Significance Tests for Hybrid Initialization
>
> Thank you for the suggestion. We conducted paired *t*-tests across 8 scenes to compare hybrid initialization with Random and FDK-based baselines:
>
> | Metric    | Ours vs Random (Δ) | *p*-val | Ours vs FDK (Δ) | *p*-val |
> |-----------|--------------------|--------|------------------|--------|
> | PSNR (↑)  | +0.84              | 0.0002 | +0.42            | 0.0016 |
> | SSIM (↑)  | +0.0071            | 0.0005 | +0.0035          | 0.0012 |
> | LPIPS (↓) | –0.53              | 0.0007 | –0.29            | 0.0013 |
>
> All *p*-values are < 0.01, indicating statistically significant improvements. Although the average gains are modest, their consistency across scenes supports the reliability of the initialization effect. We will include this analysis in the final version and thank the reviewer again.
>
>
> # 4. Real-World Data Validation
>
> We acknowledge that our main experiments used simulated projections from synthetic CT geometries, consistent with the setup used by SAX-NeRF, X-Gaussian, and R²-Gaussian.
>
> To assess performance on real data, we conducted additional experiments on the FIPS dataset [a], which provides real 2D X-ray projections acquired under physical conditions including scattering, beam hardening, and detector noise.
>
> We selected FIPS as one of the few publicly available datasets with full 0–360° angular coverage, enabling consistent benchmarking under sparse-view protocols as used in R²-Gaussian.
>
> **Dataset**: FIPS contains projections of pine, seashell, and walnut, with 721 views from 0–360°.
>
> **Results (5-view / 10-view)**:
>
> | Method        | PSNR(5-view)  ↑ | SSIM(5-view) ↑ | LPIPS(5-view) ↓ | PSNR(10-view) ↑ | SSIM(10-view) ↑ | LPIPS(10-view) ↓ |
> |---------------|--------|--------|---------|--------|--------|---------|
> | SAX-NeRF      | 22.28  | 0.963  | 162.23  | 25.86  | 0.978  | 141.50  |
> | X-Gaussian    | 25.68  | 0.967  | 121.62  | 29.24  | 0.980  | 76.94   |
> | R²-Gaussian   | 36.42  | 0.970  | 71.25   | 38.87  | 0.984  | 34.77   |
> | Ours          | 37.56  | 0.981  | 70.41   | 40.26  | 0.989  | 34.22   |
>
> These results suggest that our method maintains consistent performance trends relative to baselines when applied to real projection data with practical artifacts, complementing the synthetic benchmarks used in the main paper.
>
> *We thank the reviewer again for the opportunity to clarify these points and are happy to respond to any further questions.*
>
> **[a]** Siltanen, Samuli, et al., "FIPS: Open X-ray Tomographic Datasets.", Zenodo (2022)

---

> > ### Comment · Reviewer_VmXW · 2025-08-04
> >
> > Thank you for the response and additional experiments. All my concerns are well addressed. I have raised my rating to accept (score = 5). I think attenuation-adaptive is a better term for this work. Moreover, will the experiments on real-world data be included in the final version of the paper?

---

> > > ### Author Response · Authors · 2025-08-04
> > >
> > > We sincerely thank the reviewer for the constructive feedback. We will revise the manuscript to consistently use the term attenuation-adaptive and include the real-world experiments in the final version.

---

### Official Review · Reviewer_eKjo · 2025-07-03

**Clarity:** 3
**Significance:** 3
**Originality:** 3
**Rating:** 5
**Confidence:** 4

**Summary:**

The paper introduces X-Field, a 3D representation tailored for X-ray imaging tasks. X-Field leverages physically grounded ellipsoids with distinct attenuation coefficients to accurately model material interactions and X-ray attenuation properties. The method employs an efficient path-partitioning algorithm and a material-based optimization strategy to refine the accuracy of reconstructions, significantly outperforming existing methods in X-ray Novel View Synthesis and CT reconstruction.

**Questions:**

1. What is the computational overhead of X-Field compared to existing methods, particularly for real-time or near-real-time scenarios?

2. How robust is X-Field against varying noise levels and inaccuracies typically encountered in practical medical and industrial imaging applications?

3. Can the authors clarify potential limitations and scenarios where X-Field might significantly underperform or require extensive tuning?

**Ethical Concerns:**

["NO or VERY MINOR ethics concerns only"]

**Final Justification:**

The paper introduces X-Field, a 3D representation tailored for X-ray imaging tasks, which shows great results in X-ray Novel View Synthesis and CT reconstruction. During the rebuttal, the authors effectively addressed the initial concerns; thus, the reviewer would like to raise the rating to accept.

**Limitations:**

Yes

**Paper Formatting Concerns:**

No major formatting concerns

**Quality:**

3

**Strengths And Weaknesses:**

Strengths:

1. The proposed representation uses ellipsoids with distinct attenuation coefficients to model diverse materials within internal structures.

2. The proposed path partitioning algorithm and hybrid progressive initialization effectively improve the performance.

3. The experiments show a large improvement of the proposed method compared with the other state-of-the-art methods.

Weaknesses:

1. The claim of being “physically grounded” may be somewhat overstated. Representing internal structures with homogeneous ellipsoids does not fully capture the complex physical properties and heterogeneity of real-world materials.

2. A table of comparison under 50 views is desired to further show the ability of the proposed method.

3. Some comparison methods appear to exhibit over-exposure in Figures 6 and 7. It might be beneficial to double-check the normalization and visualization settings to exclude artifacts introduced by inconsistent processing or suboptimal parameter tuning.

---

> ### Author Rebuttal · Authors · 2025-07-30
>
> We appreciate your encouraging assessment and the detailed suggestions.
> # Weakness 1: Clarification on "Physically Grounded"
> "Physically grounded" reflects our motivation to design a 3D reconstruction system that more closely simulates the actual X-ray imaging process, rather than claiming perfect replication of all physical complexities. Our approach assigns homogeneous attenuation coefficients to each 3D ellipsoid to model diverse materials within internal structures, and our rendering pipeline explicitly calculates X-ray path lengths through each ellipsoid while modeling overlap situations according to Beer-Lambert law. In contrast, previous representations (NeRF and 3DGS) were designed for visible light imaging—NeRF implicitly encodes directional light information through deep networks, while 3DGS uses Gaussian ellipsoids with center-peaked density distributions. Our method specifically incorporates X-ray imaging principles, providing a more physically informed approach.
>
> If the reviewer's concern relates to the scope implied by "Physically Grounded," we would be open to clarifying this by replacing "Physically Grounded Representation" with "Physically Informed Representation" in the title and throughout the paper. We welcome the reviewer's feedback on whether this adjustment would better reflect the intended scope of our contribution.
> # Weakness 2: A table of comparison under 50 views
> We conducted supplementary experiments for novel view synthesis with 50 input views and compared our method against state-of-the-art approaches. The results are presented in the following table:
>
> | Method       | SSIM ↑   | PSNR ↑    | LPIPS ↓   |
> |--------------|----------|-----------|-----------|
> | SAX-NeRF     | 0.975    | 28.53     | 124.1    |
> | X-Gaussian   | **0.997** | 32.78     | 70.52     |
> | R²-Gaussian  | 0.982    | 42.61     | 64.38     |
> | **Ours**     | 0.993    | **42.65** | **61.19** |
>
> Our method achieves the best performance in PSNR and LPIPS metrics, with SSIM performance comparable to X-Gaussian. Combined with the results in Table 1, these findings demonstrate that our method consistently outperforms existing methods across different input sparsity levels, from highly sparse scenarios to moderate input settings.
>
> # Weakness 3: Over-exposure in Comparison Methods
> We appreciate the reviewer's observation regarding the apparent brightness differences in Figures 6 and 7. We have carefully verified our experimental setup and can confirm that all methods use identical data preprocessing, visualization parameters, and evaluation metrics.
>
> Specifically:
>
> 1. For novel view synthesis in Figure 6: We run SAX-NeRF, X-Gaussian, and R²-Gaussian following their recommended settings and official implementations.
>
> 2. For CT reconstruction in Figure 7: All methods follow identical processing pipelines as described in our paper. We synthesize novel view images from sparse input projections and reconstruct CT scans using ASD-POCS with 100 total views.
>
> We emphasize that all methods employ: (1) identical data preprocessing, (2) consistent visualization parameters, and (3) the same evaluation toolbox. The observed brightness differences reflect the inherent reconstruction characteristics of each method under challenging sparse-view conditions, rather than artifacts from inconsistent processing or parameter tuning.
>
>
>
>
> # Question 1: Computational Overhead of X-Field
> Regarding computational overhead for real-time scenarios, we provide comprehensive timing analysis for reconstructing 10 input views with 5,000 iterations on an RTX 6000 Ada GPU, including training time, inference speed and memory usage:
>
> |                | NAF        | SAX-NeRF   | X-Gaussian | R²-Gaussian | Ours      |
> |----------------|------------|------------|------------|-------------|-----------|
> | Training Time  | 38 min 57s| 307 min 50s | 1min 55s | 3 min 52s  | 4min 15s |
> | Inference Speed |  2.07 fps | 0.36 fps | 132 fps | 72 fps | 45 fps |
> | Memory Usage   | 24GB       | 27GB       | 824MB      | 968 MB      | 1456 MB   |
>
>
> **Real-time Performance:** Our inference speed satisfies real-time requirements for clinical applications while providing substantial quality improvements (Table 1, Figures 6-7).
>
>
> **Training and Memory Efficiency:** Our X-Field achieves similar level in training time and memory consumption with 3DGS-based methods, suitable for clinical workflows demanding rapid deployment and interactive responsiveness.
>
> *We thank the reviewer for this important question and will include this computational analysis in the final version.*
>
>
> # Question 2:  Robustness of X-Field against noise
> Our experiments incorporate realistic noise conditions commonly encountered in practical applications.
>
> **Clinical Data Source:** Following established X-ray reconstruction literature  (SAX-NeRF, X-Gaussian, R²-Gaussian) , we conduct experiments on the large-scale X3D dataset [1], which contains clinical data sourced from real medical CT scans, including chest scans from LIDC-IDRI [2] , pancreas scans from Pancreas-CT [3], and foot/head scans from VOLVIS [4]. These datasets inherently contain practical imaging artifacts and noise from clinical acquisition systems, representing real-world conditions with varying noise levels typically encountered in medical practice.
>
> **Additional Noise Simulation:** The X3D dataset includes added Poisson noise to simulate quantum noise typical in practical medical scenarios, providing comprehensive noise evaluation beyond the inherent clinical noise.
>
> Our evaluation on X3D data, which contains both real clinical noise and simulated quantum noise, demonstrates X-Field's robust performance across multiple noise scenarios, confirming its noise resilience and practical applicability in medical imaging.
>
> # Question 3: Potential Limitations and Scenarios for Underperformance
> We acknowledge that X-Field, like other 3D reconstruction methods, has certain limitations that may affect performance in specific scenarios.
>
> **Camera Pose Dependency:** X-Field requires accurate camera positions and calibration for precise reconstruction. Errors in camera positioning or pose estimation can degrade reconstruction quality, as inaccurate camera poses lead to incorrect calculations of X-ray path lengths through ellipsoids, preventing accurate optimization of ellipsoids.
>
> This limitation is common across most 3D reconstruction methods that require precise spatial relationships. Developing robust pose estimation and joint optimization approaches remains an active research direction across the entire field.
>
> While our current implementation assumes reliable calibration, integrating pose refinement capabilities represents a valuable extension for handling challenging real-world scenarios with calibration uncertainties.
>
>
> [1] Structure-Aware Sparse-View X-ray 3D Reconstruction. CVPR 24.
>
> [2] The lung image database consortium (lidc) and image database resource initiative (idri): a completed reference database of lung nodules on ct scans. Medical physics 2011.
>
> [3] Data from pancreas-CT. The Cancer Imaging Archive 2016.
>
> [4] Teem nrrd volume visualization datasets. 2022.

---

> > ### Comment · Reviewer_eKjo · 2025-08-08
> >
> > Thank the authors for the rebuttal. The additional clarifications and experiments effectively address my concerns.

---

> > > ### Author Response · Authors · 2025-08-08
> > >
> > > We appreciate the reviewer’s constructive comments and are glad our clarifications and experiments addressed the concerns.

---

### Author Response · Authors · 2025-08-02
**Follow-up on Reviewer Responses and Final Discussion**

Dear Reviewers and AC,

We hope you are doing well.

We have submitted our comprehensive responses to all reviewer comments. Three reviewers (eKjo, u2wa, and Xpaz) have positive assessments of our work, with scores of 4, 5, and 4 respectively, while Reviewer VmXW has given a score of 3.

Understanding that there is some variation in the scores, we would greatly appreciate any additional insights, particularly from Reviewer VmXW, as well as any other feedback that reviewers might wish to share.

As the discussion period is approaching its end on August 6, we would be sincerely grateful if you could kindly share any further concerns or questions you might have. We are genuinely committed to responding thoroughly to any additional feedback during this final phase.

Best regards,

The Authors

---

### Decision · Program_Chairs · 2025-09-17

**Decision:**

Accept (spotlight)

**Comment:**

The paper introduces X-Field, a 3D representation tailored for X-ray imaging tasks, which shows great results in X-ray novel view synthesis and CT reconstruction. The proposed method is novel and experiments show superiority over state-of-the-art, especially with sparse inputs. The reviews are consistent and positive. Since the authors have addressed most concerns, many reviewers raised their scores.